# Phenotypic variability and trait-specific selection in *Aegle marmelos* Correa genotypes based on morphological and quality traits

**A. K. Singh[1], Vikas Yadav[1], Lalu Prasad Yadav[1]\*, K. Gangadhara[1], Anil Pawar[1], Jagadish Rane[2], P. Ravat[1], A. Sahil[1], Prashant Kaushik[3], Ali Khadivi[4]\*, Yazgan Tunç[5]**

**1** ICAR- Central Horticultural Experiment Station (CIAH R.S.,) Godhra, Gujarat, India, **2** ICAR-Central Institute for Arid Horticulture, Bikaner, Rajasthan, India, **3** Chaudhary Charan Singh Haryana Agricultural University, Hisar, Haryana, India, **4** Department of Horticultural Sciences, Faculty of Agriculture and Natural Resources, Arak University, Arak, Iran, **5** Republic of Türkiye, Ministry of Agriculture and Forestry, General Directorate of Agricultural Research and Policies, Hatay Olive Research Institute Directorate, Hassa, Hatay, Türkiye

\* yadavlaluprasad682@gmail.com (LPY), a-khadivi@araku.ac.ir (AK)

## Abstract

This study aimed to assess the genetic variability in *Aegle marmelos* Correa to develop trait-specific genotypes based on morphological and qualitative traits. The evaluation focused on both morphological and qualitative characteristics within the gene pool of this species. High phenotypic coefficient of variation (PCV) and genotypic coefficient of variation (GCV) were observed for traits such as shell weight, fruit weight, and pulp weight, indicating substantial genetic diversity and strong potential for selective breeding within the germplasm. Heritability estimates ranged widely, with fruit weight showing a low 0.07% and shell weight a high 92.23%, reflecting the significant impact of environmental factors on trait expression. Principal Component Analysis (PCA) revealed that the first principal component (PC1) explained 40.19% of the total variation, with an eigenvalue of 8.12. The first six principal components collectively accounted for 80.77% of total variability. Genotypes CHESB-25 and CHESB-29 exhibited the highest positive PC scores for PC1 and PC2, identifying them as superior selections. Cluster analysis identified six distinct clusters of genotypes, with Cluster V being the largest and Cluster VI the smallest. This clustering highlights the genetic diversity among the bael genotypes and provides a basis for breeding and selection strategies. Cluster IV emerged as the most promising, consistently showing the highest values for key attributes such as shell weight, fruit weight, and fruit yield per plant. Therefore, prioritizing Cluster IV is recommended for selecting superior varieties and developing new cultivars. The study also noted that fruit yield per plant positively correlated with traits like shell weight and fruit weight, emphasizing the importance of these traits for yield improvement. Conversely, negative correlations with seed percent, shell percent, and phenolic content suggest these traits may be less beneficial for enhancing

**Data availability statement:** All relevant data are within the paper and its Supporting Information files.

**Funding:** The author(s) received no specific funding for this work.

**Competing interests:** The authors have declared that no competing interests exist.

yield. The hierarchical clustering heat map of the 101 bael germplasms offers a detailed perspective on the relationships between various traits and germplasms. The results offer vital information for creating A. marmelos cultivars with higher yields and better quality. For breeding programs, targeted selection is made possible by the discovery of important clusters and superior genotypes (CHESB-25 and CHESB-29). Given the high level of genetic variation found, hybridization may be able to improve desired characteristics like fruit output and weight. Overall, the findings offer important insights for selecting elite genotypes and advancing breeding programs.

## Introduction

The bael tree (*Aegle marmelos* Correa), an ancient species from India, belongs to the Rutaceae family. Though often underappreciated, it has been known since prehistoric times and holds a revered place in Hindu mythology. Hindus consider this tree sacred, offering its fruits and leaves to deities such as Goddess Parvati and Lord Shiva. Its medicinal value is well-documented in ancient Indian texts, including *Upvana Vinod*, the Sanskrit *Charaka Samhita*, and *Brihat Samhita*. The fruit is also mentioned in the Yajur Veda (Om Prakash, 1961). Bael's rich history of therapeutic use is further illustrated by its depiction in *Ajanta* cave paintings and references in Buddhist and Jain literature [1–3]. Keeping the above facts in the background, a large number of germplasms were collected and established in the field gene bank of the station clonally.

Bael thrives in areas with poor water sources and offers a wealth of functional vitamins and nutrients. As the global demand for superfoods and natural antioxidants rises, bael presents a significant opportunity for developing innovative food products [4]. It is used to create a range of value-added items, including squash, murabba, fruit slabs, toffees, powders, and jams [5–7]. The ripe pulp of premium bael cultivars, as well as the "sherbat" made from it, is highly valued for its mild laxative, tonic, digestive, and restorative properties, and is effective in treating biliousness. Conversely, green, unripe bael fruits are commonly used to prepare "murabba," a preserve renowned for its digestive benefits. In India, a popular drink called "sherbat" is made by blending seeded bael pulp with milk or water and sugar [8,9]. Although bael fruits are rarely consumed as a dessert due to their astringent taste, numerous seeds, and high fiber content, the pulp contains marmelosin, which imparts restorative, astringent, and laxative benefits [10]. All parts of the bael tree—stem, bark, root, leaf, flower, seed oil, and fruit at any stage of maturity—are used in Ayurvedic medicine. The fruit and leaves are known for their proven antioxidant, cardiotonic, antifungal, and analgesic properties. Bael is naturally found in the sub-Himalayan region of India, as well as in dry deciduous forests in central and southern India [11]. Various landraces of bael are distributed across a wide range of environments, including tropical, arid, semi-arid, and dry subtropical zones. Bael genotypes are notably prevalent in the plains of Gujarat, Rajasthan, and Uttar Pradesh, thriving particularly in arid climates, hilly areas, woodlands, and

wastelands [12,13]. To identify superior variants, it is essential to evaluate a diverse range of genotypes under rainfed semi-arid conditions, as significant variation in both qualitative and quantitative traits is observed among them [14,15].

In traditional Indian medicine, the root, bark, leaves, flowers, and fruit of the bael tree are used for their ethnopharmacological properties, including anti-diabetic, antiviral, antifungal, and anticancer effects [16,17]. Bioactive compounds isolated from these parts show potential for treating various human diseases and conditions [18,19]. Bael leaf poultices are applied to ulcers and eye infections, while leaf powder serves as a repellent against storage pests. The bark and mature fruit are utilized in the tanning and dyeing industries, and the leaves are employed in treating beriberi and dropsy [20,21]. Diluted leaf extracts provide relief from tartar, and research has explored the potential of leaf alkaloids in treating asthma [22,23]. The young leaves and tender shoots are used as animal fodder, and the gum from the stem and seed locules is used to stabilize drilling fluids. Additionally, bael wood is used for making small agricultural tools and as pulp for wrapping paper [24,25].

The use of morpho-pomological and biochemical profiling allows for a comprehensive evaluation of genetic diversity and agronomic traits in bael, which is critical for the conservation of its germplasm and the planning of an effective breeding program [21].

In this investigation, we analyzed the morphometric, growth, yield, and quality diversity within the bael gene pool to identify elite genotypes specifically adapted to rainfed semi-arid conditions in western India. Our efforts focused on characterizing various traits to pinpoint superior genotypes suited for these challenging environmental conditions. The study aimed to uncover genetic variations that could enhance bael cultivation under rainfed conditions. By evaluating these traits, we sought to identify and promote the best-performing bael genotypes for improved crop performance in semi-arid regions. To enhance variety quality in all horticultural aspects, the presence of genetic variability within a population is essential for the planning and execution of effective crop improvement programs [15]. Greater variability in crop plants offers an opportunity to identify the genotypes having desirable traits to fulfill the needs of the stakeholders [19]. Understanding the genetic diversity within bael germplasm is key to advancing crop improvement efforts and developing targeted breeding programs.

## Materials and methods

The present study was carried out at the Bael Germplasm Experimental Block of the National Active Germplasm Site, situated at the Central Horticultural Experiment Station (ICAR-CIAH) in Godhra, India 2022–2024. The location is at latitude 22°41'38" N, longitude 73°33'22" E, and an elevation of 113 meters above sea level. This station maintains the world's largest germplasms of bael (217 clonal and 121 seedlings). The collection sites of bael genotypes have been presented in Fig 1.

Budded bael trees were raised on seedling rootstocks developed from seeds of a single plant, spaced at 7 m × 6 m. The age of the trees in the orchard is 12 years. Each genotype was represented by six plants, with two plants serving as a replication. The planting layout adhered to a randomized block design. The experimental site was characterized by loam to clay loam having151.25, 8.22 and 143.50 kg/ha available nitrogen, phosphorus, and potassium, respectively, at 0–15 cm soil depth with pH 7.50, EC 1.14dS/m, bulk density1.42g/cc, hydraulic conductivity0.28 cm/h and organic carbon 0.33%. During the experiment, a uniform dose of 50 kg of farmyard manure /tree was applied and cultivated purely under rainfed conditions in Gujarat. The experimental site is characterized by a semi-arid, hot climate with an average annual precipitation of 750 mm and significant temperature fluctuations, ranging from 35–46°C in summer to 9–25°C in winter [26,27]. The standard package of practices was followed according to Singh et al. [21].

The samples of marketable fruits were harvested from pest and disease-free healthy plants for different observations and analyses. The samples were washed with tap water and excess water was drained. Fresh samples of fruit were used for the determination of Total soluble solids (TSS) and acidity. TSS of fruits and mucilage were determined using a digital refractometer and results were expressed as °Brix [25,28]. The pulp acidity contents of the samples

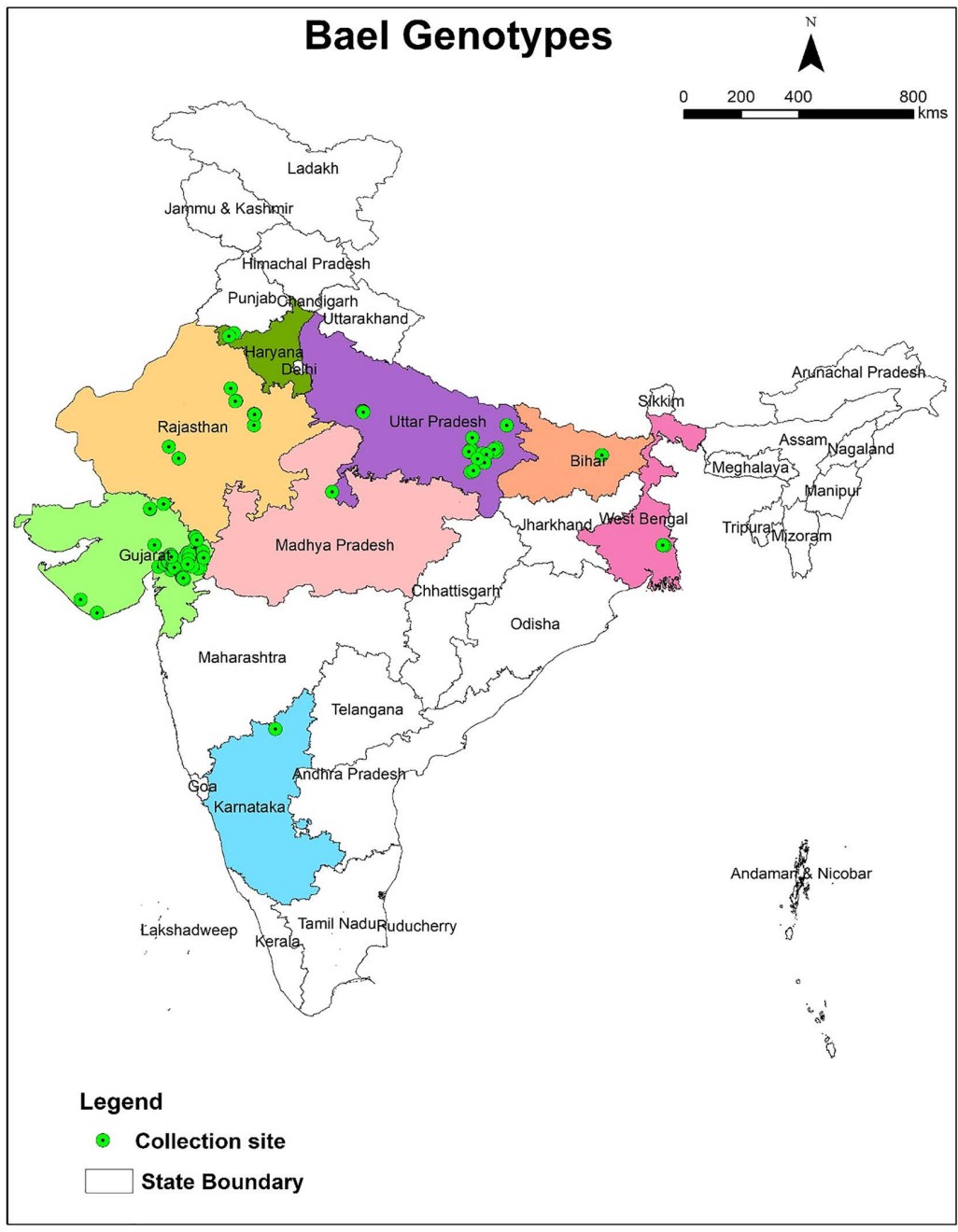

**Fig 1. Collection sites of bael genotypes developed using ArcGIS (version 10.8) software.** The map was generated by the third author, Lalu Prasad Yadav, using ArcGIS software v10.1 and is original; no third-party copyrighted or proprietary material was used, and no permission is required. (ArcGIS: http://www.arcgis.com/home/item.html?id=30e5fe3149c34df1ba922e6f5bbf808f).

were determined by visual titration with 0.1 N sodium hydroxide (NaOH) to an endpoint of pH 8.1 as suggested by Ranganna (1986).

Total phenolics were estimated using the Folin–Ciocalteu reagent [Singleton et al. 1999]. Results were expressed as gallic acid equivalent (mg GAE/100 g fresh weight, FW). Ascorbic acid content was determined in accordance with the dinitrophenylhydrazine (DNPH) method. The value was expressed as mg/100 g FW [27,28].

**Ethics approval and consent to participate**: Not Applicable

**Research involving Human Participants and, or Animals:** Not applicable

### Statement on experimental research and field studies on plants

The either cultivated or wild-growing plants sampled comply with relevant institutional, national, and international guidelines and domestic legislation of India.

### Data analysis

Burton's [29] method is followed in the computation of the variance components and coefficients of variation. One derived expected general genetic progress and heritability using the formulae Johnson et al. [30]. Al-Jibouri et al. [31] developed a technique for evaluating all possible feature combinations by assessing correlation coefficients at both phenotypic and genotypic levels, focusing on variances in these coefficients. RStudio (version 2023.3.0+) tools were used for Data processing and graphical visualization. Principal component analysis (PCA) and facetted density plots were performed using the most current available versions of the FactoMineR, factoextra, and ggplot2 packages, drawing on fundamental methods by Hotelling [32] and Pearson (1901). Cluster analysis was conducted with the cluster, factoextra, dendextend, and ggplot2 programs following Sokal and Michener [33]. Variance analysis used the variability and agriculture packages, while correlation analysis was conducted with the corrplot package [Posit 34]. Additionally, ordinary least squares (OLS) regression analysis was employed to investigate the relationship between the selected characteristics and fruit yield per plant. The results from this regression analysis highlight the significance of these characteristics in predicting yield outcomes, as they offer a model that explains a substantial portion of the variance in fruit yield per plant.

## Results and discussion

### Morphological variations among *genotypes*

The evaluation of the *Aegle marmelos* gene pool under rainfed semi-arid conditions is significant for several reasons. It provides empirical evidence on how different genotypes perform regarding growth, yield, and quality traits in challenging environments. This research is crucial for identifying suitable varieties and elite genotypes that can thrive in arid and semi-arid conditions, which is essential for effective crop improvement and adaptation strategies. Observations of prominent morphometric variability in the *A. marmelos* germplasm—such as differences in growth patterns, fruit characteristics, and quality attributes—are illustrated through various Figs. These include variability in fruit and growth traits (Figs 2–4), genetic diversity in fruit components (Figs 5–6), and variations in locule arrangement, seed characteristics, and fiber color (Figs 7–9). Additionally, variations in fruit size and shape, thorn orientation, stigma color, and stem bark color (Figs 10–13) highlight the diverse potential within the gene pool. Raw data of the studied germplasm of bael is shown in S1 File. This detailed documentation underscores the importance of understanding these variations for selecting and breeding elite genotypes suited for semi-arid conditions.

### Genetic variability studies

Estimates of genetic parameters for various yield traits in bael are presented in Table 1. High phenotypic coefficient of variation (PCV) and genotypic coefficient of variation (GCV) were observed for shell weight (27.29 and 27.21), fruit

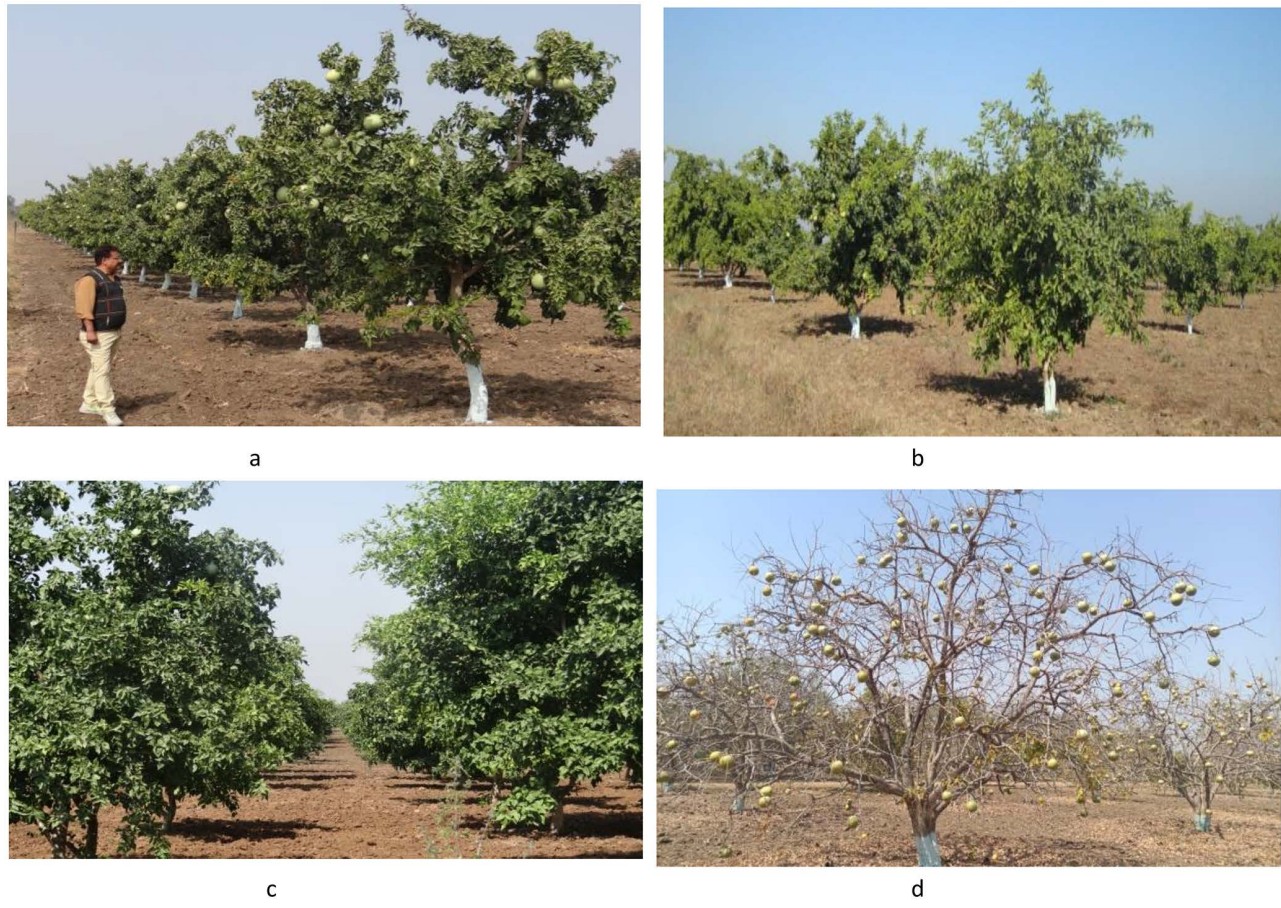

**Fig 2. Different bael genotypes at field repository of ICAR-CHES, Godhra, Gujarat, India.** The above images were taken by the third author, Lalu Prasad Yadav, and are original; no third-party copyrighted or proprietary material was used, and no permission is required.

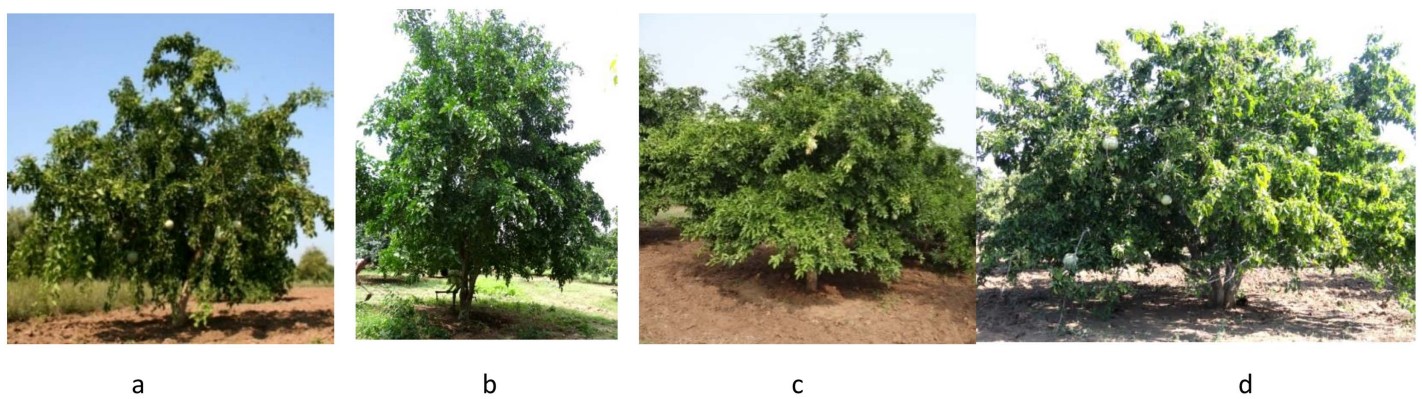

**Fig 3. Growth behavior of bael genotypes: a) drooping, b) erect, c) semi-spreading, and d) spreading.** The above images were taken by the third author, Lalu Prasad Yadav, and are original; no third-party copyrighted or proprietary material was used, and no permission is required.

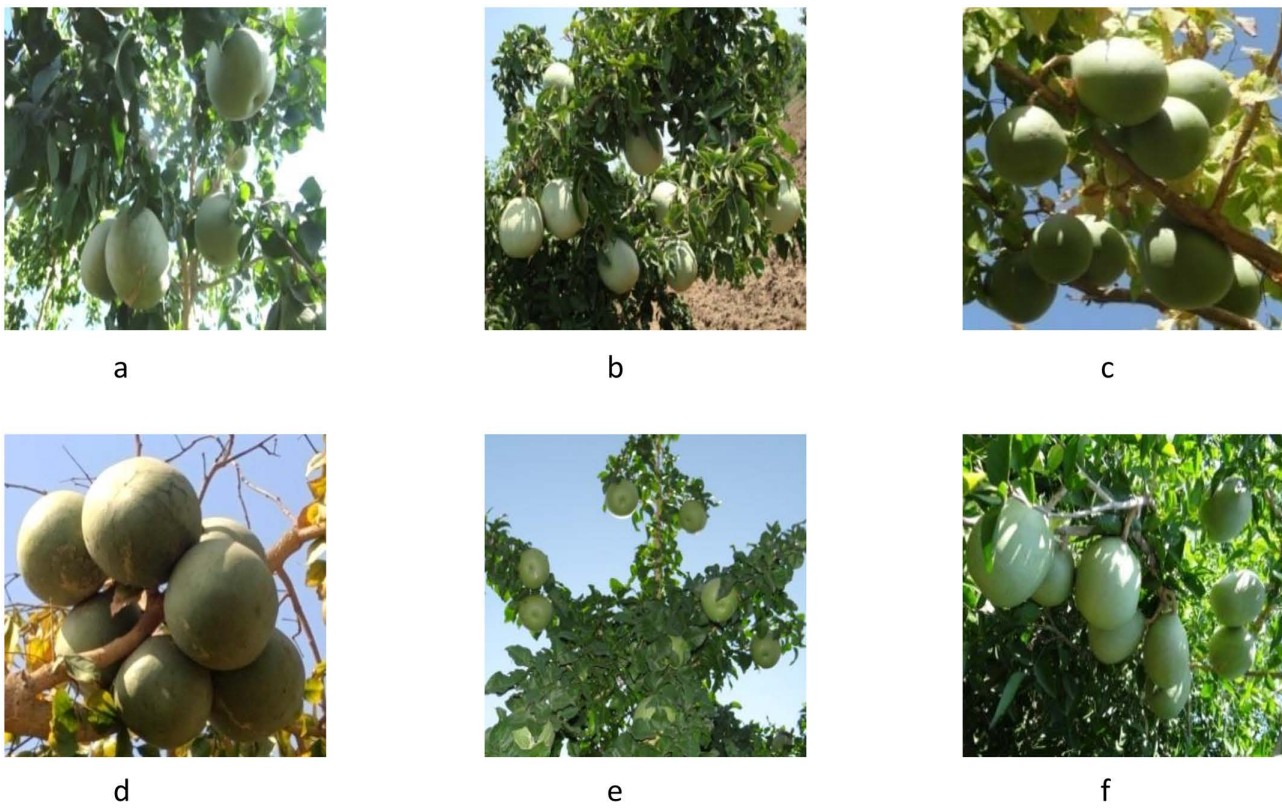

**Fig 4. Bearing behavior variation in fruit of different bael genotypes: a) CHESB-11, b) CHESB-1, c) CHESB-8, d) CHESB-27, e) CHESB-31, and f) CHESB-16.** The above images were taken by the third author, Lalu Prasad Yadav, and are original; no third-party copyrighted or proprietary material was used, and no permission is required.

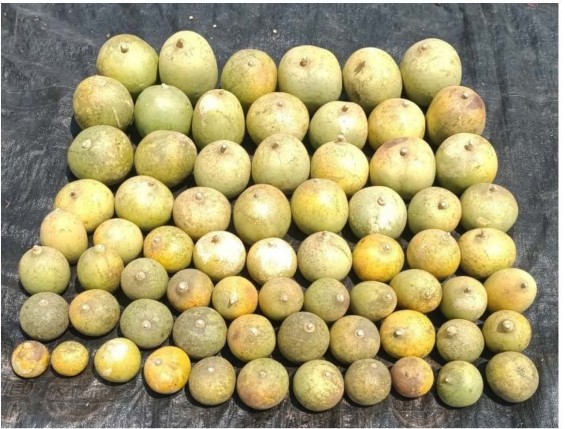 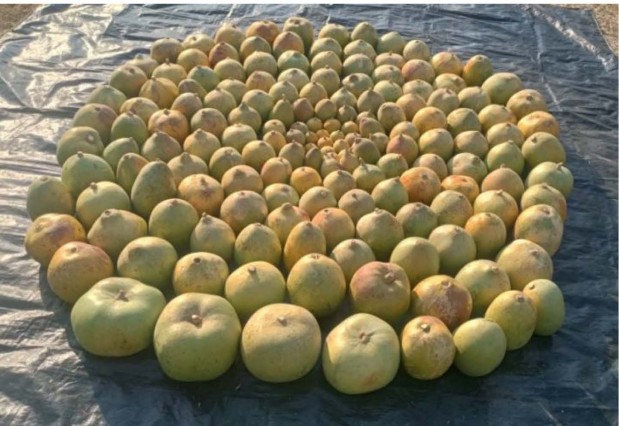

**Fig 5. Variability in fruit shape and size of bael genotypes.** The above images were taken by the third author, Lalu Prasad Yadav, and are original; no third-party copyrighted or proprietary material was used, and no permission is required.

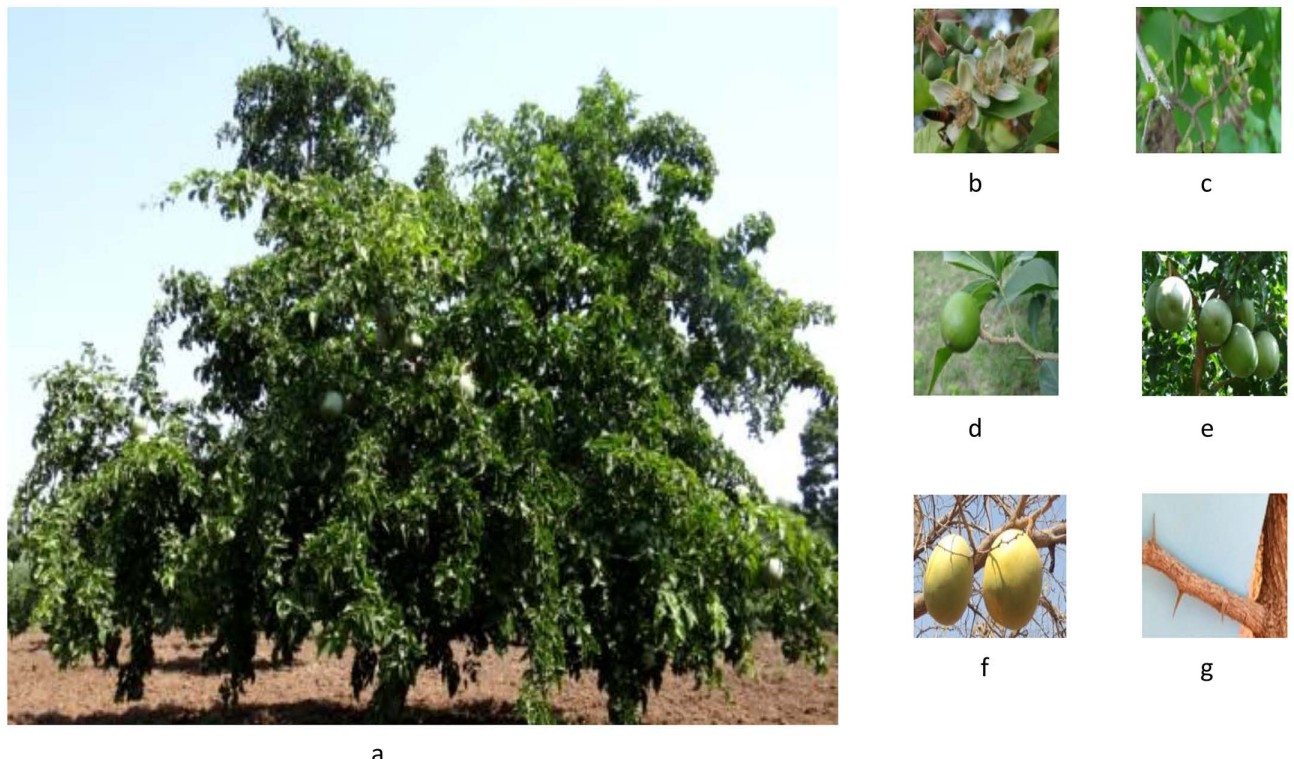

**Fig 6. Representative photographs of bael genotype conserved at field gene bank ICAR-CIAH-CHES, Godhra, Gujarat: a) full-grown tree, b) flower and pollinator, c) fruit set, d) developing fruit, e) full-developed fruit, f) ripened fruit, and g) thorn size.** The above images were taken by the third author, Lalu Prasad Yadav, and are original; no third-party copyrighted or proprietary material was used, and no permission is required.

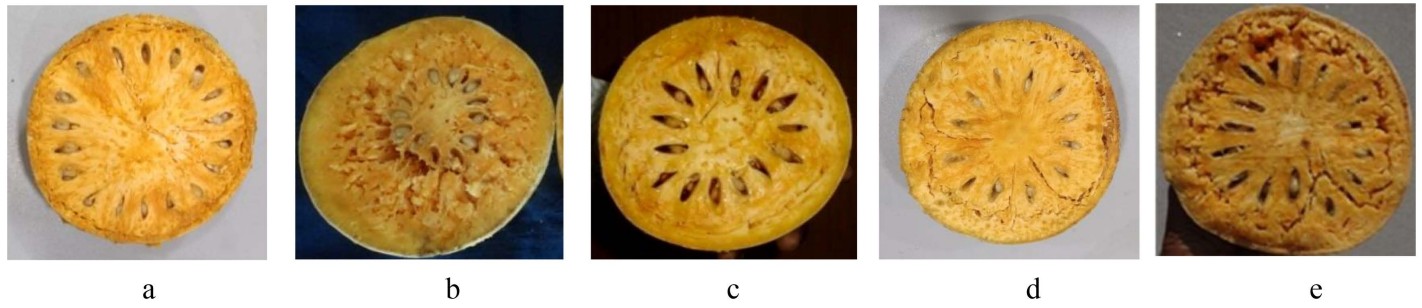

**Fig 7. Variation in locule arrangement of different bael genotypes: a) peripheral, b) highly centric, c) centric, d) semi-centric, and e) scattered.** The above images were taken by the third author, Lalu Prasad Yadav, and are original; no third-party copyrighted or proprietary material was used, and no permission is required.

weight (30.27 and 30.14), pulp weight (33.83 and 33.79), shell thickness (20.16 and 20.12), number of seeds per fruit (29.26 and 29.24), fiber weight (29.00 and 28.21), seed percentage (30.89 and 30.74), shell percentage (22.71 and 22.68), and phenolic content (20.99 and 20.82). These high PCV and GCV values indicate substantial variability within the germplasm, suggesting strong potential for selection [35,36,37]. Moderate PCV and GCV were found for fruit circumference (16.99 and 16.83), total seed weight (17.05 and 17.01), vitamin C (17.44 and 17.41), non-reducing

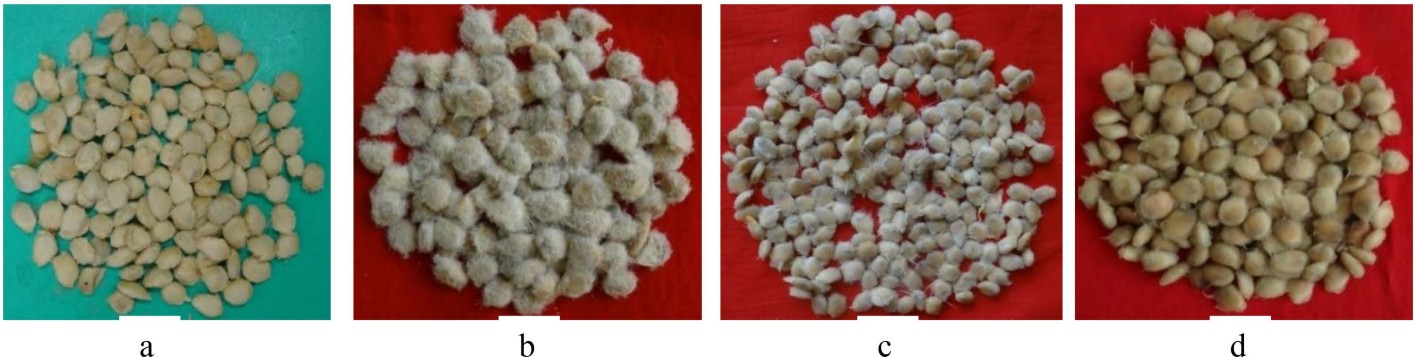

**Fig 8. Variation in seed size, shape, and fiber of different bael genotypes: a) small seed and fiberless, b) bold seed with fiber, c) small seed with fiber, and d) bold seed and fiberless.** The above images were taken by the third author, Lalu Prasad Yadav, and are original; no third-party copyrighted or proprietary material was used, and no permission is required.

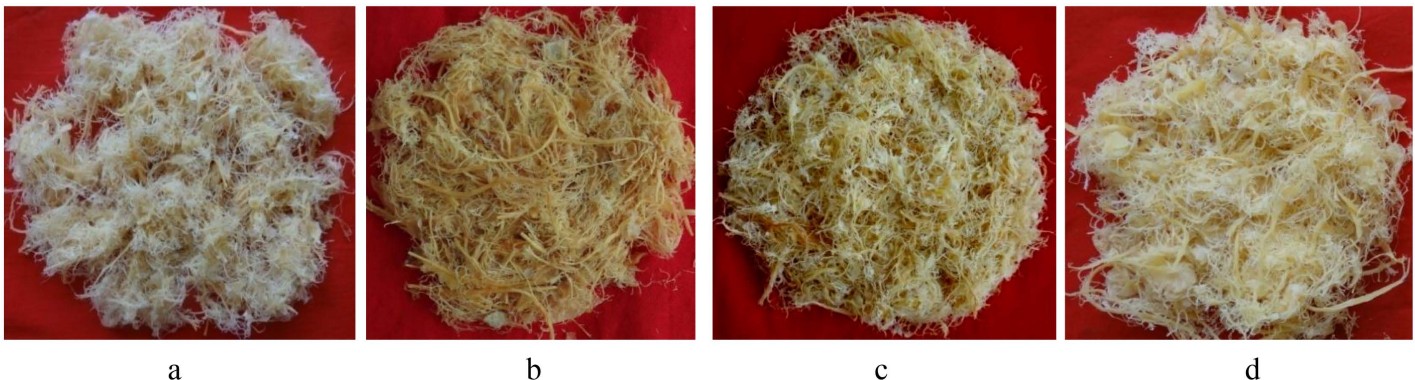

**Fig 9. Variation in fiber color of different bael genotypes: a) whitish-yellow, b) yellow, c) light yellow, and d) yellowish-white.** The above images were taken by the third author, Lalu Prasad Yadav, and are original; no third-party copyrighted or proprietary material was used, and no permission is required.

sugar (10.08 and 10.01), fruit length (16.24 and 16.21), fruit width (16.99 and 16.86), and fruit yield per plant (13.13 and 13.11), indicating a moderate level of genetic variability in these traits. In contrast, low PCV and GCV values were recorded for pulp percentage (5.69 and 5.57) and TSS pulp (7.92 and 7.87), reflecting minimal variability within the evaluated germplasm for these traits [38,39,40].

High broad-sense heritability supports the identification of favorable traits for selection, allowing breeders to focus on superior genotypes based on their phenotypic expression of quantitative traits. Heritability ranged from 0.07% for fruit weight to 92.23% for shell weight, indicating that environmental factors largely influence these traits. Genetic advances as a percentage of the mean ranged from 0.39% for total sugar to 25.07% for fiber weight. Notably, high genetic advances were observed for fiber weight (25.07%), while moderate advances were noted for shell weight (11.92%) and number of seeds per fruit (17.99%). High heritability coupled with low genetic advances as a percentage of the mean for shell weight suggests non-additive gene action and limited selection response. Conversely, low heritability and genetic advances for fruit weight (0.07% and 1.36%), pulp weight (0.06% and 1.68%), pulp percentage (7.05% and 0.50%), shell thickness (0.10% and 0.56%), fruit circumference (14.98% and 6.27%), fruit width (1.75% and 2.32%), fruit length (2.17% and 2.32%), total seed weight (7.16% and 4.51%), seed percentage (0.14% and 2.00%), shell percentage (3.98% and 5.98%), TSS pulp (2.99% and 0.61%), vitamin C (3.74% and 3.47%), acidity (10.04% and

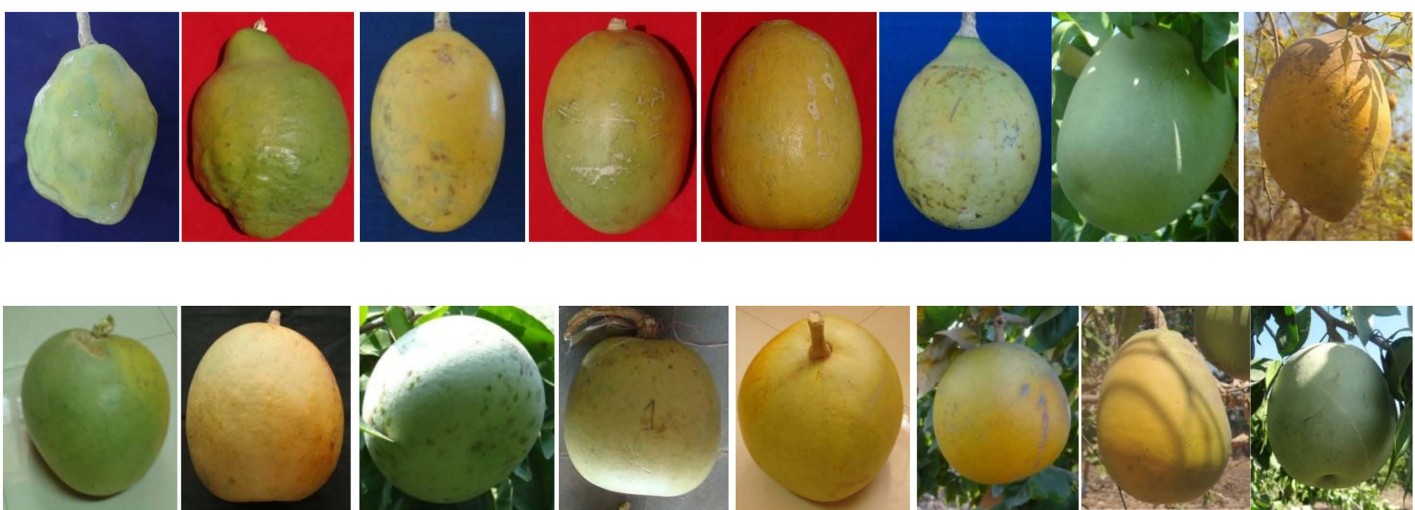

**Fig 10. Variation in fruit size and shape of different bael genotypes under rainfed semi-arid conditions.** The above images were taken by the third author, Lalu Prasad Yadav, and are original; no third-party copyrighted or proprietary material was used, and no permission is required.

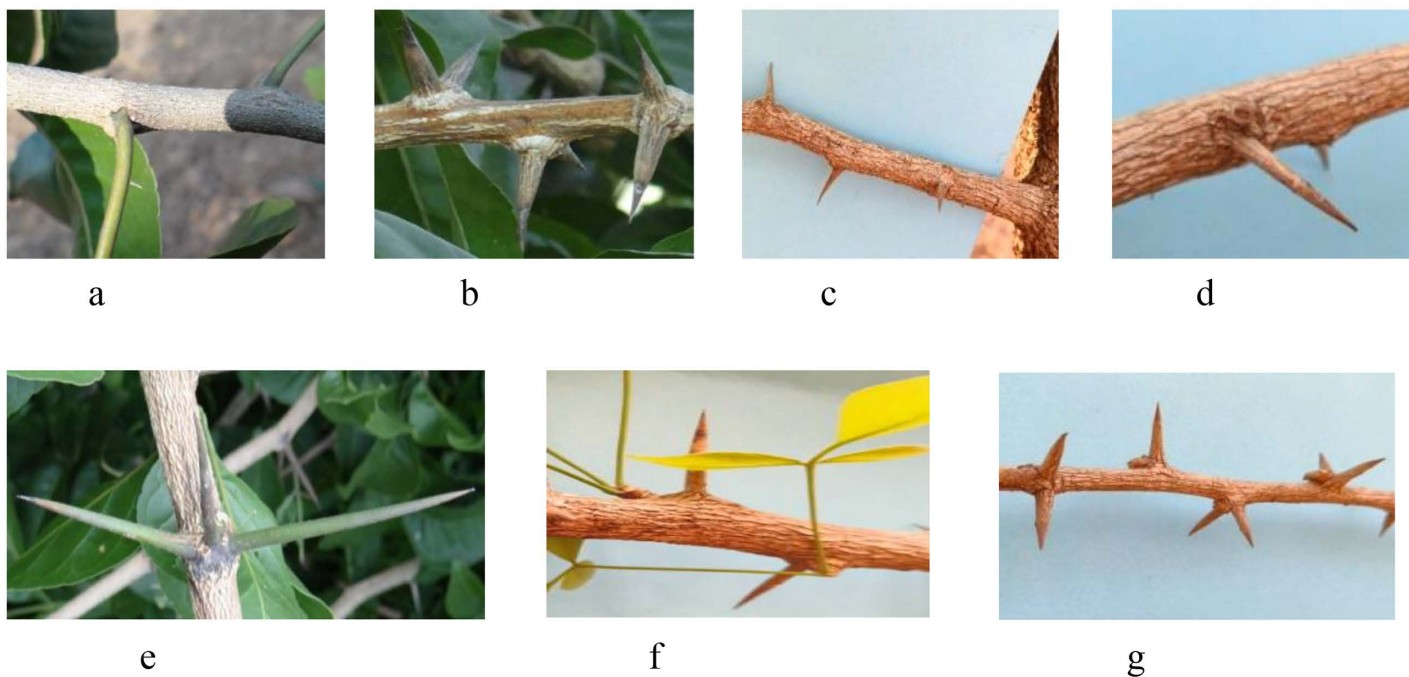

a b c d

e f g

**Fig 11. Variation in thorn orientation size and shape of different bael genotypes: a) thornless, b) double at node, medium, and stout, c) single, small, and thin, d) single, long, and stout, e) three at node, long, and medium thin, f) single at node, medium, and stout, and g) double at node, small, bold, and stout.** The above images were taken by the third author, Lalu Prasad Yadav, and are original; no third-party copyrighted or proprietary material was used, and no permission is required.

6.83%), total sugar (0.95% and 0.39%), phenolic content (11.07% and 8.32%), and non-reducing sugar (1.09% and 0.64%) indicate that these traits are highly influenced by environmental factors and are less responsive to selection [41,42,40,43,44].

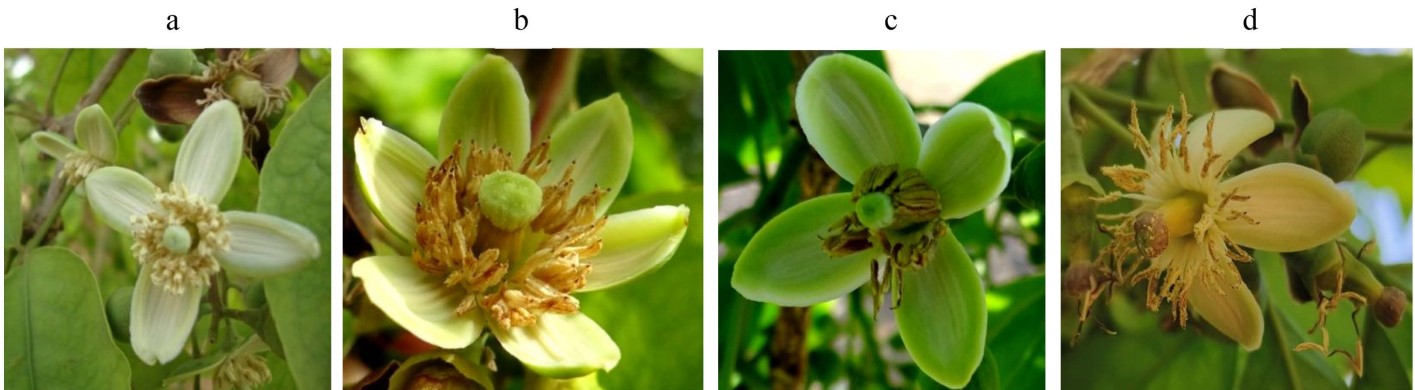

**Fig 12. Variation in stigma color in bael genotypes: a) greenish-white, b) light green, c) green, and d) purple.** The above images were taken by the third author, Lalu Prasad Yadav, and are original; no third-party copyrighted or proprietary material was used, and no permission is required.

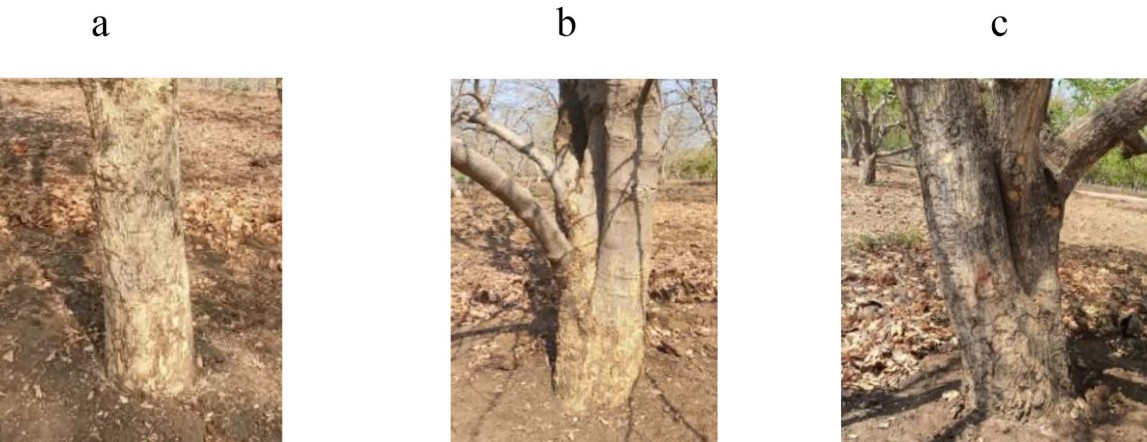

**Fig 13. Variation in stem bark color of bael genotypes: a) grey-brown, b) yellow-grey, and c) brown-grey.** The above images were taken by the third author, Lalu Prasad Yadav, and are original; no third-party copyrighted or proprietary material was used, and no permission is required.

## Character association studies

The genotypic correlation coefficients among various characters of bael genotypes, as detailed in Table 2 and Fig 14 reveal several key insights. Fruit yield per plant demonstrated positive correlations with shell weight (0.55), fruit weight (0.78), pulp weight (0.77), pulp percentage (0.65), shell thickness (0.31), fruit circumference (0.70), fruit width (0.70), fruit length (0.67), number of seeds per fruit (0.46), total seed weight (0.39), fiber weight (0.11), TSS pulp (0.01), vitamin C (0.09), acidity (0.06), and total sugar (0.01). Thus, these traits should be prioritized to enhance yield per plant. Conversely, negative correlations were observed with seed percentage (−0.73), shell percentage (−0.52), phenolic content (−0.08), and non-reducing sugar (−0.01) [3,45]. Non-reducing sugar positively correlated with pulp percentage (0.04), shell thickness (0.10), number of seeds per fruit (0.02), fiber weight (0.15), TSS pulp (0.01), total sugar (0.91), and phenolic content (0.20), but negatively with shell weight (−0.05), fruit weight (−0.01), pulp weight (−0.01), fruit circumference (−0.09), fruit width (−0.09), fruit length (−0.004), total seed weight (−0.04), shell percentage (−0.04), seed percentage (−0.06), vitamin C (−0.21), and acidity (−0.004). Phenolic content was positively associated with pulp percentage (0.04), shell thickness

**Table 1. Estimates of genetic parameters for various characters in bael genotypes.**

| No. | Character | Unit | GCV | PCV | h² | GA | GAM |
|---|---|---|---|---|---|---|---|
| 1 | Shell weight | g | 27.21 | 27.29 | 92.23 | 25.17 | 11.92 |
| 2 | Fruit weight | kg | 30.14 | 30.27 | 0.07 | 0.02 | 1.36 |
| 3 | Pulp weight | kg | 33.79 | 33.83 | 0.06 | 0.02 | 1.68 |
| 4 | Pulp | % | 5.57 | 5.69 | 7.05 | 0.40 | 0.50 |
| 5 | Shell thickness | mm | 20.12 | 20.16 | 0.10 | 0.02 | 0.56 |
| 6 | Fruit circumference | cm | 16.83 | 16.99 | 14.98 | 2.55 | 6.27 |
| 7 | Fruit width | cm | 16.86 | 16.99 | 1.75 | 0.30 | 2.32 |
| 8 | Fruit length | cm | 16.21 | 16.24 | 2.17 | 0.35 | 2.32 |
| 9 | No. of seeds/fruit | Number | 29.24 | 29.26 | 86.19 | 25.22 | 17.99 |
| 10 | Total seed weight | g | 17.01 | 17.05 | 7.16 | 1.22 | 4.51 |
| 11 | Fiber weight | g | 28.21 | 29.00 | 36.06 | 10.46 | 25.07 |
| 12 | Seed | % | 30.74 | 30.89 | 0.14 | 0.04 | 2.00 |
| 13 | Shell | % | 22.68 | 22.71 | 3.98 | 0.90 | 5.98 |
| 14 | TSS pulp | °Brix | 7.52 | 7.58 | 2.99 | 0.23 | 0.61 |
| 15 | Vitamin C | mg/100 g | 17.41 | 17.44 | 3.74 | 0.65 | 3.47 |
| 16 | Acidity | % | 5.17 | 5.22 | 10.04 | 0.05 | 6.83 |
| 17 | Total sugar | % | 7.87 | 7.92 | 0.95 | 0.08 | 0.39 |
| 18 | Phenolic content | mg/g | 20.82 | 20.99 | 11.07 | 2.32 | 8.32 |
| 19 | Non reducing sugar | mg/g | 10.01 | 10.08 | 1.09 | 0.11 | 0.64 |
| 20 | Fruit yield/plant | kg | 13.11 | 13.13 | 36.94 | 4.85 | 5.05 |

GCV- Genotypic co-efficient of variation, PCV- Phenotypic co-efficient of variation, h²- Heritability (broad sense), GA- Genetic advance, GAM- Genetic advance as % mean.

(0.03), vitamin C (0.08), acidity (0.05), and total sugar (0.22), while negatively correlated with shell weight (−0.12), fruit weight (−0.10), pulp weight (−0.09), fruit circumference (−0.24), fruit width (−0.24), fruit length (−0.15), number of seeds per fruit (−0.16), total seed weight (−0.27), fiber weight (−0.07), shell percentage (−0.04), seed percentage (−0.08), and TSS pulp (−0.06). Total sugar showed positive correlations with pulp percentage (0.003), shell thickness (0.03), fruit length (0.002), fiber weight (0.05), TSS pulp (0.04), and acidity (0.01), but negative correlations with shell weight (−0.05), fruit weight (−0.04), pulp weight (−0.03), fruit circumference (−0.06), fruit width (−0.06), number of seeds per fruit (−0.09), total seed weight (−0.17), shell percentage (−0.01), seed percentage (−0.10), and vitamin C (−0.15) [3,46,47].

Acidity was positively correlated with fruit weight (0.01), pulp weight (0.01), fruit circumference (0.01), fruit width (0.01), fruit length (0.03), number of seeds per fruit (0.06), total seed weight (0.05), TSS pulp (0.08), and vitamin C (0.14), but negatively with shell weight (−0.02), shell thickness (−0.03), fiber weight (−0.03), seed percentage (−0.003), and shell percentage (−0.04). Vitamin C exhibited positive associations with fruit weight (0.01), pulp weight (0.03), pulp percentage (0.21), fruit length (0.004), and TSS pulp (0.16), while negatively correlated with shell weight (−0.13), shell thickness (−0.14), fruit circumference (−0.01), fruit width (−0.01), number of seeds per fruit (−0.05), total seed weight (−0.01), fiber weight (−0.15), seed percentage (−0.02), and shell percentage (−0.20). TSS pulp showed positive correlations with shell weight (0.08), fruit weight (0.04), pulp weight (0.04), shell thickness (0.01), fruit circumference (0.18), fruit width (0.18), fruit length (0.12), number of seeds per fruit (0.14), total seed weight (0.22), seed percentage (0.10), and shell percentage (0.06), but negative correlations with pulp percentage (−0.01) and fiber weight (−0.14). Shell percentage was positively correlated with shell weight (0.17), shell thickness (0.02), and seed percentage (0.56), but negatively with fruit weight (−0.53), pulp weight (−0.61), pulp percentage (−0.95), fruit circumference (−0.41), fruit width (−0.41), fruit length (−0.42), number of seeds per fruit (−0.27), total seed weight (−0.24), and fiber weight (−0.02). Seed

**Table 2. Genotypic correlation between morphological, yield, and fruit quality characters in bael genotypes.**

| Trait | 1 | 2 | 3 | 4 | 5 | 6 | 7 | 8 | 9 | 10 | 11 | 12 | 13 | 14 | 15 | 16 | 17 | 18 | 19 | 20 |
|---|---|---|---|---|---|---|---|---|---|---|---|---|---|---|---|---|---|---|---|---|
| 1 | 1 | 0.72** | 0.64** | 0.07 | 0.59** | 0.65** | 0.65** | 0.64** | 0.50** | 0.45** | 0.08 | -0.48 | 0.17* | 0.08 | -0.13 | -0.02 | -0.05 | -0.12 | -0.05 | 0.55** |
| 2 | | 1 | 0.99** | 0.68** | 0.53** | 0.86** | 0.86** | 0.85** | 0.67** | 0.60** | 0.06 | -0.71 | -0.53 | 0.04 | 0.01 | 0.01 | -0.04 | -0.10 | -0.01 | 0.78** |
| 3 | | | 1 | 0.74** | 0.49** | 0.85** | 0.85** | 0.84** | 0.66** | 0.58** | 0.03 | -0.72 | -0.61 | 0.04 | 0.03 | 0.01 | -0.03 | -0.09 | -0.01 | 0.77** |
| 4 | | | | 1 | 0.07 | 0.57** | 0.57** | 0.57** | 0.34** | 0.28** | -0.15 | -0.70 | -0.95 | -0.01 | 0.21** | 0.05 | 0.003 | 0.040 | 0.01 | 0.65** |
| 5 | | | | | 1 | 0.33** | 0.33** | 0.34** | 0.44** | 0.34** | 0.16 | -0.28 | 0.02 | 0.01 | -0.14 | -0.03 | 0.03 | 0.03 | 0.10 | 0.31** |
| 6 | | | | | | 1 | 0.98** | 0.92** | 0.60** | 0.59** | -0.02 | -0.58 | -0.41 | 0.18 | -0.01 | 0.01 | -0.06 | -0.24 | -0.09 | 0.70** |
| 7 | | | | | | | 1 | 0.92** | 0.60** | 0.59** | -0.02 | -0.58 | -0.41 | 0.18 | -0.01 | 0.01 | -0.06 | -0.24 | -0.09 | 0.70** |
| 8 | | | | | | | | 1 | 0.59** | 0.55** | 0.01 | -0.61 | -0.42 | 0.12 | 0.004 | 0.03 | 0.002 | -0.15 | -0.004 | 0.67** |
| 9 | | | | | | | | | 1 | 0.89** | 0.06 | -0.12 | -0.27 | 0.14 | -0.05 | 0.06 | -0.09 | -0.16 | 0.02 | 0.46** |
| 10 | | | | | | | | | | 1 | 0.10 | 0.03 | -0.24 | 0.22* | -0.01 | 0.05 | -0.17 | -0.27 | -0.04 | 0.39** |
| 11 | | | | | | | | | | | 1 | -0.09 | 0.02 | -0.14 | -0.15 | -0.03 | 0.05 | -0.07 | 0.15* | 0.11 |
| 12 | | | | | | | | | | | | 1 | 0.56** | 0.10 | -0.02 | -0.003 | -0.10 | -0.08 | -0.06 | -0.73 |
| 13 | | | | | | | | | | | | | 1 | 0.06 | -0.20 | -0.04 | -0.01 | -0.04 | -0.04 | -0.52 |
| 14 | | | | | | | | | | | | | | 1 | 0.16* | 0.08 | 0.04 | -0.06 | 0.03 | 0.01 |
| 15 | | | | | | | | | | | | | | | 1 | 0.14 | -0.15 | 0.08 | -0.21 | 0.09 |
| 16 | | | | | | | | | | | | | | | | 1 | 0.01 | 0.05 | -0.004 | 0.06 |
| 17 | | | | | | | | | | | | | | | | | 1 | 0.22** | 0.91** | 0.01 |
| 18 | | | | | | | | | | | | | | | | | | 1 | 0.20* | -0.08 |
| 19 | | | | | | | | | | | | | | | | | | | 1 | -0.01 |
| 20 | | | | | | | | | | | | | | | | | | | | 1 |

For trait numbers, please see Table 1.

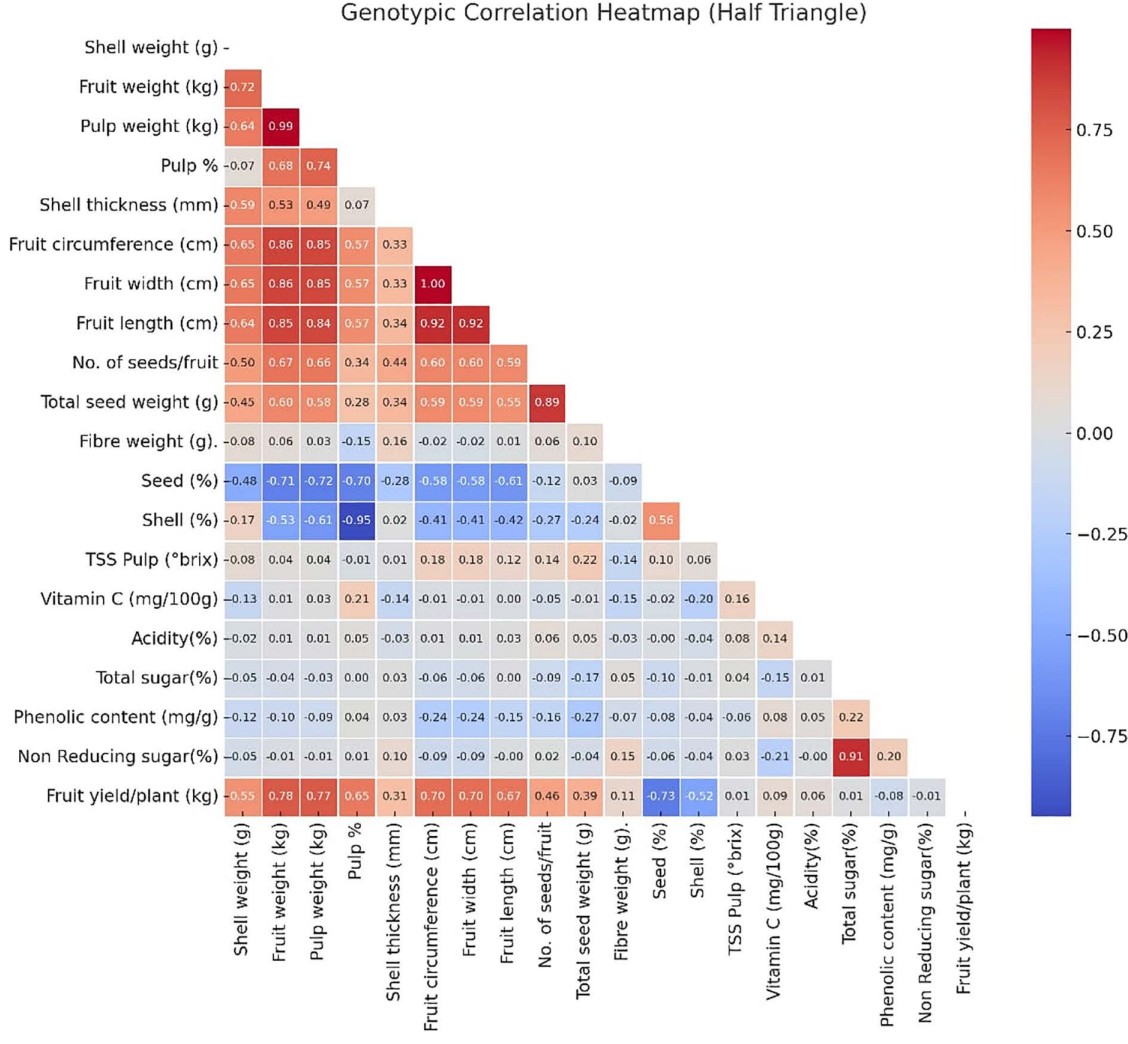

**Fig 14. Genotypic correlation coefficient heatmap between 20 yield and quality characters in bael genotypes.**

percentage positively correlated only with total seed weight (0.03), but negatively with shell weight (−0.48), fruit weight (−0.71), pulp weight (−0.72), pulp percentage (−0.70), shell thickness (−0.28), fruit circumference (−0.58), fruit width (−0.58), fruit length (−0.61), number of seeds per fruit (−0.12), and fiber weight (−0.09). Fiber weight showed positive correlations with shell weight (0.08), fruit weight (0.06), pulp weight (0.03), shell thickness (0.16), fruit length (0.01), number of seeds per fruit (0.06), and total seed weight (0.10), but negative correlations with pulp percentage (−0.15), fruit circumference (−0.02), and fruit width (−0.02). Total seed weight was positively associated with shell weight (0.45), fruit weight (0.60), pulp weight (0.58), pulp percentage (0.28), shell thickness (0.34), fruit circumference (0.44), fruit width (0.60), fruit length (0.60), and number of seeds per fruit (0.59). The number of seeds per fruit correlated positively with shell weight (0.50), fruit weight (0.67), pulp weight (0.66), pulp percentage (0.34), shell thickness (0.44), fruit circumference (0.60), fruit width (0.60), and fruit length (0.59). Fruit length and fruit width exhibited positive associations with shell weight (0.64 and 0.65), fruit weight (0.85 and 0.86), pulp weight (0.84 and 0.85), pulp percentage (0.57 and 0.57) and shell thickness (0.33 and 0.33). Finally, fruit circumference showed positive correlations with shell weight (0.65), fruit weight (0.86), pulp weight (0.85), pulp percentage (0.57) and shell thickness (0.33), while shell thickness

positively correlated with shell weight (0.59), fruit weight (0.53), pulp weight (0.49) and pulp percentage (0.07). The pulp percent showed positive associations with shell weight (0.07), fruit weight (0.68), and pulp weight (0.74) at genotypic levels. Pulp weight exhibited positive correlations with both shell weight (0.64) and fruit weight (0.99) at genotypic levels. The fruit weight showed a positive association with shell weight (0.72) at genotypic levels [3,48,44,49].

Overall, the most significant traits for improving fruit yield per plant are shell weight, fruit weight, pulp weight, pulp percentage, shell thickness, fruit circumference, fruit width, and fruit length. Additionally, quality parameters like TSS pulp, vitamin C, acidity, and total sugar are also important for enhancing fruit quality [3,50,51,49].

## Principal component analysis

The eigenvalues and their percentage variations are detailed in Table 3 and Fig 15. As the number of principal components (PCs) increases, the eigenvalue and variance associated with each principal component decrease, while the

Table 3. Eigenvalues, percent variability, and cumulative percent variability for various principal components in bael genotypes.

| Principal components | Eigenvalue | Percentage of variance (%) | Cumulative percentage of variance (%) |
|---|---|---|---|
| PC-1 | 8.12 | 40.19 | 40.19 |
| PC-2 | 2.32 | 11.49 | 51.68 |
| PC-3 | 2.14 | 10.60 | 62.29 |
| PC-4 | 1.47 | 7.25 | 69.54 |
| PC-5 | 1.19 | 5.88 | 75.42 |
| PC-6 | 1.08 | 5.36 | 80.78 |
| PC-7 | 0.94 | 4.65 | 85.43 |
| PC-8 | 0.84 | 4.16 | 89.59 |
| PC-9 | 0.65 | 3.19 | 92.79 |
| PC-10 | 0.56 | 2.77 | 95.56 |
| PC-11 | 0.36 | 1.78 | 97.34 |
| PC-12 | 0.20 | 0.99 | 98.33 |

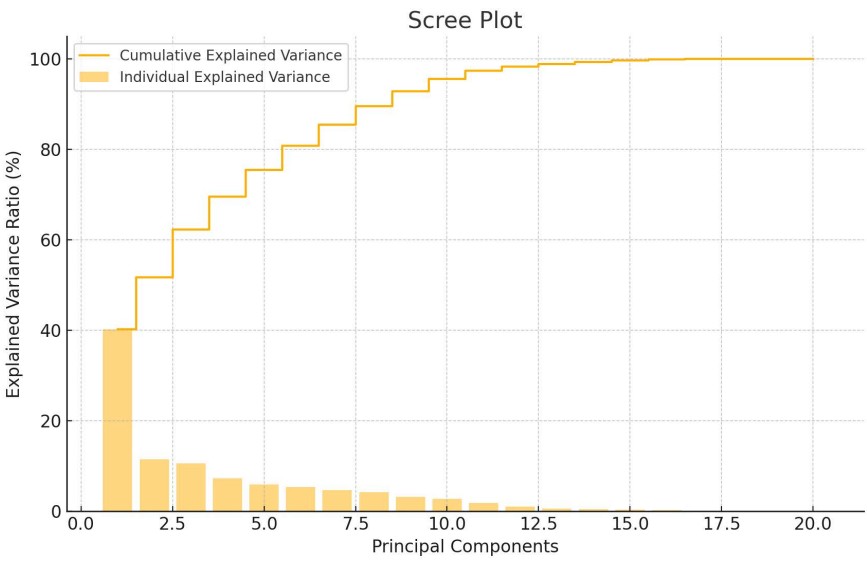

Fig 15. PCA scree plot of bael genotypes between explained variance ratio and principal components.

cumulative variability increases. The scree plot illustrates the percentage of variation explained by each PC, showing that the first principal component accounts for approximately 40.19% of the variability with the highest eigenvalue of 8.12 (Fig 15). Subsequent principal components exhibit a progressive decrease in eigenvalue: the second PC has an eigenvalue of 2.32, the third PC 2.14, the fourth PC 1.47, the fifth PC 1.19, and the sixth PC 1.08. The percentage of total variation explained by these PCs also decreases, with PC-2 explaining 11.49%, PC-3 10.60%, PC-4 7.25%, PC-5 5.88%, and PC-6 5.36%. Among the twelve PCs identified, only the first six (PC1 to PC6) have eigenvalues greater than 1.00, indicating that these components capture significant traits influencing the phenotype of the individuals. Together, they account for approximately 80.77% of the total variation observed across the 101 cultivars for the 20 traits studied [3,52,53, 43,44,54].

**The PC scores for 101 bael genotypes**

In principal component analysis (PCA), PC scores represent the projection of the original data onto the principal components. These scores quantify each variable's contribution to the principal components and help identify which variables are most influential in explaining the data's variation. Additionally, PC scores are instrumental in various analyses, such as clustering, classification, and regression, enabling the exploration of relationships between variables and principal components [3,49,52,55]. The PC scores for both the first principal component (PC1) and the second principal component (PC2) include both positive and negative values (Table 4).

For the first principal component, the highest positive PC score was recorded for CHESB-25 (146.64), followed by CHESB-29(129.25), CHESB-39(99.23), and CHESB-45 (99.05). The lowest positive score in this component was (4.74) for CHESB-13. Conversely, the negative PC scores ranged from −0.04 for CHESB-74 to −134.50 for CHESB-12. In the second principal component, the highest positive PC score was (156.65) for CHESB-25, while the lowest positive score was (0.35) for CHESB-97. The negative PC scores varied from (−0.03) for CHESB-61 to (−97.67) for CHESB-60.

The relationship between each trait and PC1 *vs.* PC2 was presented in Fig 16. In first and second principal component, CHESB-4, CHESB-13, CHESB-17, CHESB-18, CHESB-19, CHESB-20, CHESB-23, CHESB-25, CHESB-31, CHESB-37, CHESB-43, CHESB-53, CHESB-56, CHESB-58, CHESB-62, CHESB-68, CHESB-70, CHESB-72, CHESB-76, CHESB-77, CHESB-81, CHESB-83, CHESB-88, CHESB-89, CHESB-91, CHESB-93 and CHESB-96 depicted positive active genotypes. The contribution of these genotypes was greater for the variability in both PC1 and PC2.

**Hierarchical clustering analysis**

Hierarchical cluster analysis (HCA) is essential for identifying superior genotypes and tailoring breeding strategies. The analysis resulted in six clusters comprising 101 genotypes Table 5 and Fig 17. Cluster V was the largest, including 36 genotypes(CHESB-1, CHESB-4, CHESB-5, CHESB-6,CHESB-8, CHESB-11, CHESB-13, CHESB-15, CHESB-16, CHESB-17, CHESB-19, CHESB-20, CHESB-21, CHESB-25, CHESB-27, CHESB-37, CHESB-43, CHESB-46, CHESB-49, CHESB-53, CHESB-56, CHESB-58, CHESB-62, CHESB-63, CHESB-65, CHESB-66, CHESB-68, CHESB-70, CHESB-72, CHESB-74, CHESB-76, CHESB-77, CHESB-78, CHESB-88, CHESB-91, CHESB-96), followed by Cluster II with 24 genotypes(CHESB-7, CHESB-22, CHESB-24, CHESB-30, CHESB-32, CHESB-35, CHESB-38, CHESB-40, CHESB-44, CHESB-47, CHESB-54, CHESB-55, CHESB-57, CHESB-60, CHESB-80, CHESB-82, CHESB-84, CHESB-86, CHESB-89, CHESB-92, CHESB-94, CHESB-97, CHESB-99, CHESB-101), Cluster III with 19 genotypes(CHESB-3, CHESB-18, CHESB-23, CHESB-29, CHESB-34, CHESB-39, CHESB-41, CHESB-45, CHESB-50, CHESB-52, CHESB-71, CHESB-73, CHESB-79, CHESB-81, CHESB-83, CHESB-85, CHESB-93, CHESB-95, CHESB-98), and Cluster I with 12 genotypes(CHESB-2, CHESB-9, CHESB-10, CHESB-14, CHESB-26, CHESB-31, CHESB-33, CHESB-42, CHESB-48, CHESB-59, CHESB-64, CHESB-69). Cluster IV contained 9 genotypes (CHESB-12, CHESB-28,

**Table 4. The PCA scores for bael genotypes.**

| Sr. No | Genotypes | PC1 | PC2 | Sr. No | Genotypes | PC3 | PC4 |
|---|---|---|---|---|---|---|---|
| 1 | CHESB-1 | 77.08 | −36.61 | 52 | CHESB-52 | 89.98 | −34.42 |
| 2 | CHESB-2 | 22.66 | −1.20 | 53 | CHESB-53 | 50.59 | 32.06 |
| 3 | CHESB-3 | 7.31 | −12.61 | 54 | CHESB-54 | −26.69 | 5.32 |
| 4 | CHESB-4 | 29.72 | 37.58 | 55 | CHESB-55 | −62.17 | 35.37 |
| 5 | CHESB-5 | −8.69 | −19.47 | 56 | CHESB-56 | 50.81 | 17.78 |
| 6 | CHESB-6 | −48.87 | 30.62 | 57 | CHESB-57 | −54.67 | −25.26 |
| 7 | CHESB-7 | −43.44 | −26.65 | 58 | CHESB-58 | 47.47 | 45.10 |
| 8 | CHESB-8 | 53.69 | −41.42 | 59 | CHESB-59 | −23.37 | 40.76 |
| 9 | CHESB-9 | −61.97 | 4.27 | 60 | CHESB-60 | −24.77 | −97.67 |
| 10 | CHESB-10 | −59.91 | −1.64 | 61 | CHESB-61 | −115.03 | −0.03 |
| 11 | CHESB-11 | 83.54 | −49.81 | 62 | CHESB-62 | 43.25 | 12.54 |
| 12 | CHESB-12 | −134.50 | 34.76 | 63 | CHESB-63 | −14.44 | −10.03 |
| 13 | CHESB-13 | 4.74 | 19.38 | 64 | CHESB-64 | 55.24 | −4.63 |
| 14 | CHESB-14 | −15.86 | 14.97 | 65 | CHESB-65 | −5.37 | 18.63 |
| 15 | CHESB-15 | −24.24 | 5.29 | 66 | CHESB-66 | −0.90 | 10.61 |
| 16 | CHESB-16 | 29.98 | −54.71 | 67 | CHESB-67 | −123.60 | 15.79 |
| 17 | CHESB-17 | 34.05 | 59.15 | 68 | CHESB-68 | 10.04 | 28.14 |
| 18 | CHESB-18 | 46.99 | 59.94 | 69 | CHESB-69 | 30.96 | −12.29 |
| 19 | CHESB-19 | 53.48 | 35.09 | 70 | CHESB-70 | 12.96 | 25.73 |
| 20 | CHESB-20 | 44.23 | 37.85 | 71 | CHESB-71 | 80.96 | −14.92 |
| 21 | CHESB-21 | 10.34 | −35.03 | 72 | CHESB-72 | 10.06 | 4.41 |
| 22 | CHESB-22 | −61.76 | −46.94 | 73 | CHESB-73 | 79.02 | −3.31 |
| 23 | CHESB-23 | 49.01 | 40.35 | 74 | CHESB-74 | −0.04 | 5.47 |
| 24 | CHESB-24 | −45.94 | −14.65 | 75 | CHESB-75 | −122.72 | 13.41 |
| 25 | CHESB-25 | 146.64 | 156.65 | 76 | CHESB-76 | 5.59 | 25.66 |
| 26 | CHESB-26 | −57.52 | −3.89 | 77 | CHESB-77 | 45.12 | 7.11 |
| 27 | CHESB-27 | 60.35 | −61.91 | 78 | CHESB-78 | −0.14 | −13.00 |
| 28 | CHESB-28 | −116.87 | 3.03 | 79 | CHESB-79 | 90.63 | −34.41 |
| 29 | CHESB-29 | 129.25 | −22.11 | 80 | CHESB-80 | −51.48 | −33.89 |
| 30 | CHESB-30 | −54.21 | −5.76 | 81 | CHESB-81 | 53.18 | 19.00 |
| 31 | CHESB-31 | 4.82 | 1.31 | 82 | CHESB-82 | −47.50 | −32.55 |
| 32 | CHESB-32 | −39.86 | −35.34 | 83 | CHESB-83 | 74.49 | 1.06 |
| 33 | CHESB-33 | −8.55 | −10.02 | 84 | CHESB-84 | −46.70 | 1.74 |
| 34 | CHESB-34 | 82.30 | −14.10 | 85 | CHESB-85 | 82.49 | −35.89 |
| 35 | CHESB-35 | −56.28 | 2.84 | 86 | CHESB-86 | −56.56 | −11.43 |
| 36 | CHESB-36 | −121.50 | 11.67 | 87 | CHESB-87 | −125.27 | 39.87 |
| 37 | CHESB-37 | 25.46 | 45.29 | 88 | CHESB-88 | 39.57 | 16.28 |
| 38 | CHESB-38 | −36.12 | −19.41 | 89 | CHESB-89 | 15.00 | 6.65 |
| 39 | CHESB-39 | 99.23 | −42.74 | 90 | CHESB-90 | −116.87 | 9.44 |
| 40 | CHESB-40 | −55.09 | −25.13 | 91 | CHESB-91 | 26.11 | 41.30 |
| 41 | CHESB-41 | 87.89 | −29.96 | 92 | CHESB-92 | −56.48 | −25.89 |
| 42 | CHESB-42 | −21.84 | −2.67 | 93 | CHESB-93 | 56.23 | 33.21 |
| 43 | CHESB-43 | 37.77 | 42.94 | 94 | CHESB-94 | −51.00 | −5.73 |
| 44 | CHESB-44 | −54.90 | −30.14 | 95 | CHESB-95 | 84.57 | −28.59 |
| 45 | CHESB-45 | 99.05 | −43.59 | 96 | CHESB-96 | 6.12 | 16.57 |
| 46 | CHESB-46 | −22.25 | −6.54 | 97 | CHESB-97 | −38.90 | 0.35 |

*(Continued)*

**Table 4.** (Continued)

| Sr. No | Genotypes | PC1 | PC2 | Sr. No | Genotypes | PC3 | PC4 |
|---|---|---|---|---|---|---|---|
| 47 | CHESB-47 | −24.80 | 8.49 | 98 | CHESB-98 | 85.92 | −19.23 |
| 48 | CHESB-48 | −51.41 | −31.20 | 99 | CHESB-99 | −64.73 | −12.58 |
| 49 | CHESB-49 | 11.84 | −4.11 | 100 | CHESB-100 | −101.79 | 18.31 |
| 50 | CHESB-50 | 69.69 | −33.59 | 101 | CHESB-101 | −65.97 | −8.58 |
| 51 | CHESB-51 | −3.94 | 28.12 | | | | |

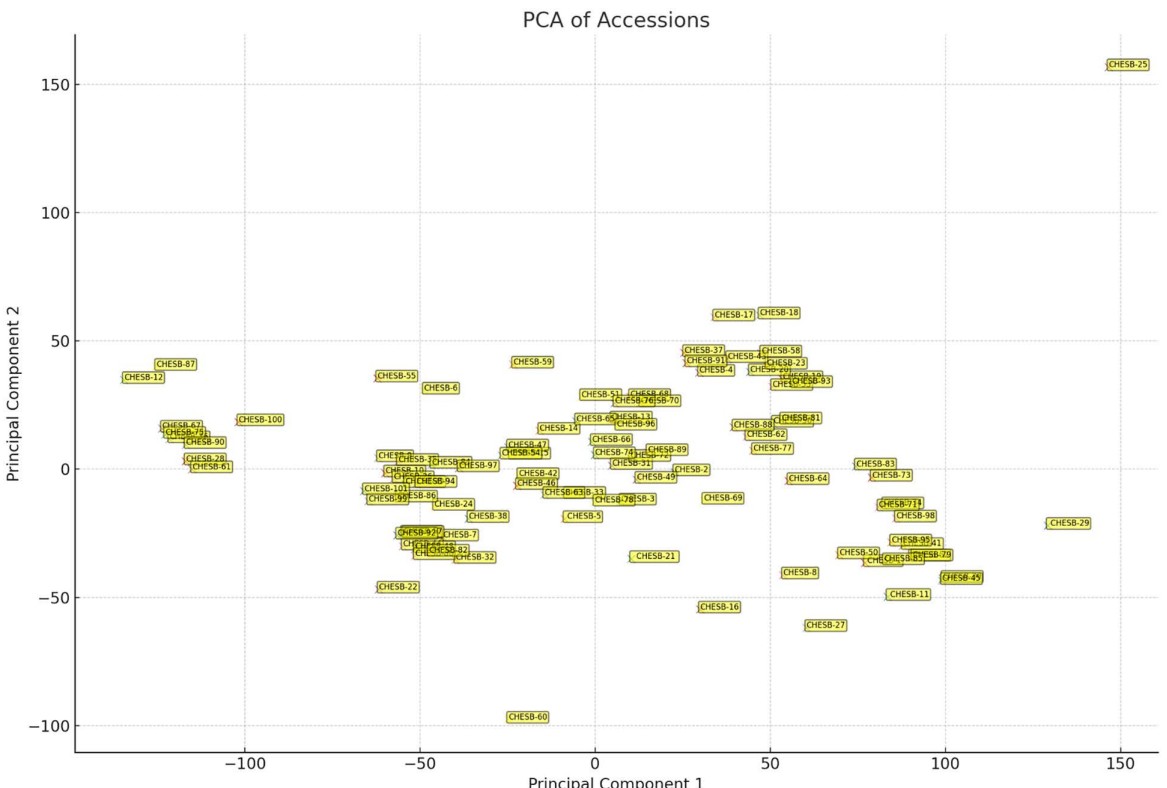

**Fig 16. Distributions of bael genotypes across two PCs.**

CHESB-36, CHESB-61, CHESB-67, CHESB-75, CHESB-87, CHESB-90, CHESB-100), while Cluster VI had only a single genotype (CHESB-51). This clustering provides valuable insights into the genetic diversity among the studied cultivars, revealing distinct groupings based on phenotypic traits and offering important implications for breeding and selection strategies (Fig 17) [3,56,57]. The relationships among the different genotypes are illustrated in the dendrogram. Genotypes clustered closely together in the dendrogram are more like one another compared with those farther apart. Additionally, the dendrogram displays the relative similarity among genotypes across the various clusters [3,43,44,52,53,54].

Hierarchical Clustering Heatmap (HCH) represents twenty-yield and quality characters of the 101bael genotype (Fig 18). Based on the Hierarchical Clustering Heatmap (HCH), five groups were created by loosely organizing the one-hundred-one *A. marmelos* germplasm. Among the five groups, group V had the maximum number of genotypes

**Table 5. Cluster composition based on D² statistics of bael genotypes in six clusters.**

| Cluster | No. of Genotype | Genotypes |
|---|---|---|
| Cluster I | 12 | CHESB-2, CHESB-9, CHESB-10, CHESB-14, CHESB-26, CHESB-31, CHESB-33, CHESB-42, CHESB-48, CHESB-59, CHESB-64, CHESB-69 |
| Cluster II | 24 | CHESB-7, CHESB-22, CHESB-24, CHESB-30, CHESB-32, CHESB-35, CHESB-38, CHESB-40, CHESB-44, CHESB-47, CHESB-54, CHESB-55, CHESB-57, CHESB-60, CHESB-80, CHESB-82, CHESB-84, CHESB-86, CHESB-89, CHESB-92, CHESB-94, CHESB-97, CHESB-99, CHESB-101 |
| Cluster III | 19 | CHESB-3, CHESB-18, CHESB-23, CHESB-29, CHESB-34, CHESB-39, CHESB-41, CHESB-45, CHESB-50, CHESB-52, CHESB-71, CHESB-73, CHESB-79, CHESB-81, CHESB-83, CHESB-85, CHESB-93, CHESB-95, CHESB-98 |
| Cluster IV | 09 | CHESB-12, CHESB-28, CHESB-36, CHESB-61, CHESB-67, CHESB-75, CHESB-87, CHESB-90, CHESB-100 |
| Cluster V | 36 | CHESB-1, CHESB-4, CHESB-5, CHESB-6,CHESB-8, CHESB-11, CHESB-13, CHESB-15, CHESB-16, CHESB-17, CHESB-19, CHESB-20, CHESB-21, CHESB-25, CHESB-27, CHESB-37, CHESB-43, CHESB-46, CHESB-49, CHESB-53, CHESB-56, CHESB-58, CHESB-62, CHESB-63, CHESB-65, CHESB-66, CHESB-68, CHESB-70, CHESB-72, CHESB-74, CHESB-76, CHESB-77, CHESB-78, CHESB-88, CHESB-91, CHESB-96 |
| Cluster VI | 01 | CHESB-51 |

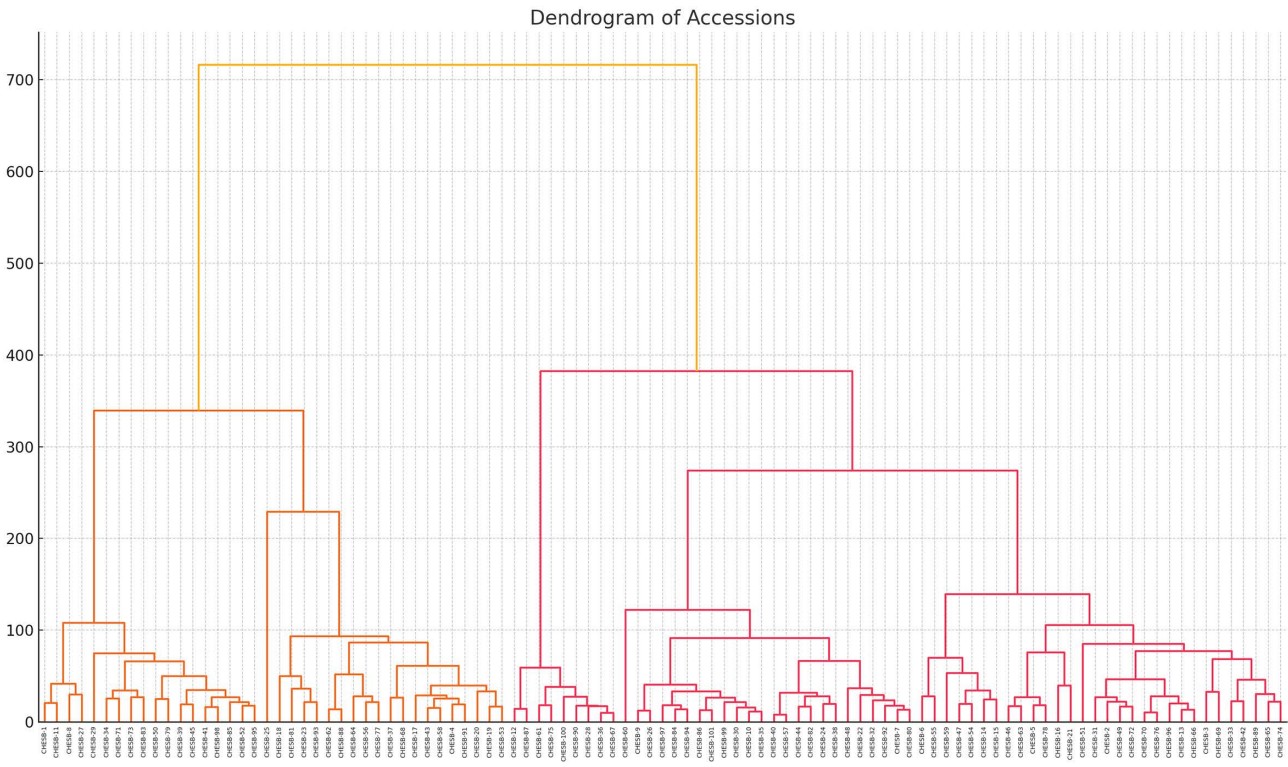

**Fig 17. Dendrogram showing the relationship among bael genotypes.**

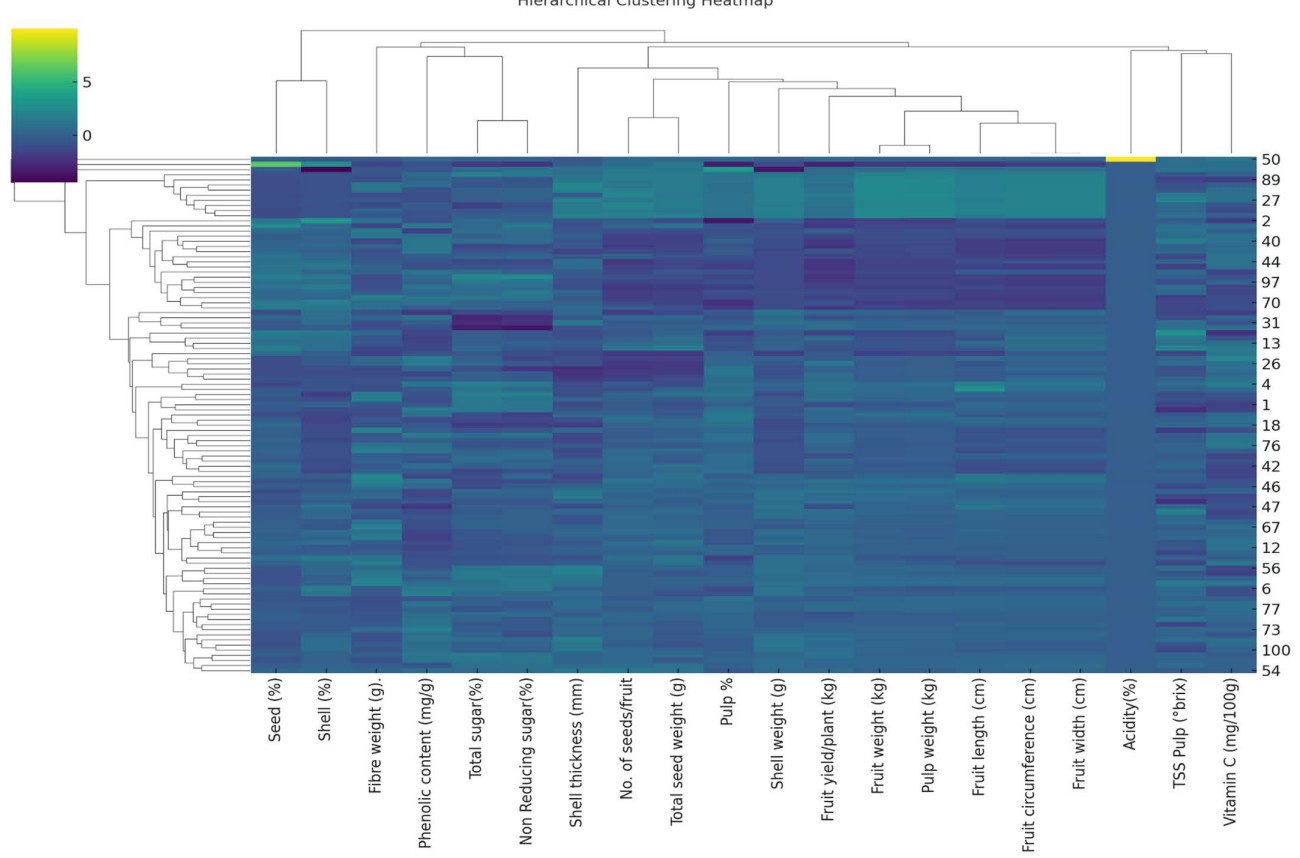

**Fig 18. Hierarchical clustering heatmap (HCH) represents 20 yield and quality characters of bael genotype.**

(71), followed by group IV (18). The lowest genotype was found in group II (1), followed by group I (2). The hierarchical clustering heat map shows the linkages and trends among numerous dataset attributes and samples. The heatmap displaying the standardized characteristics values uses color gradients; darker colors imply lower values and brighter hues suggest higher values. Whereas the left dendrogram unites similar traffic samples, the top dendrogram clusters similar traits based on their expression patterns across samples. This visualization enables one to identify groupings of characteristics with regular fluctuations and groups of samples with similar qualities. Such patterns would enable one to grasp the primary organization of the data by revealing connections and similarities that might direct environmental investigations, genetic research, and breeding plans. Analyzing the clusters and color patterns allows researchers to understand the relationships between properties and identify significant characteristics separating different samples. The acidity% in CHESB-50 is a primary trait separating high-performing germplasms from others and may be focused on further breeding and genetic studies of bael. Yield-related traits like fruit weight, pulp weight, and number of seeds per fruit directly influence the overall yield per plant, indicating better yield potential. Likewise, some quality traits, including pulp Percentage, TSS, vitamin C, and acidity, significantly impact the overall quality of bael, and physical characteristics, including shell weight and thickness, provided insights into the overall fruit structure and marketability. Studying the genetic relationships between germplasm is made more accessible by cluster analysis, which may put germplasm with comparable genetic information into one group [26,27,57] also mentioned that the two-dimensional plot generated from PCA showed three groups that were similar to the clustering pattern of the UPGMA dendrogram.

## Cluster mean analysis

Clusters mean analysis exhibits ample variation for all the growth and yield attributing characters and were presented (Table 6). The shell weight of the fruit exhibited the highest cluster mean value in Cluster IV (304.51), followed by Cluster II (256.37), with the lowest value recorded in Cluster III (149.42). For fruit weight, the maximum cluster mean of 2.48 was observed in Cluster IV, followed by Cluster II at 1.62, while the minimum mean was noted in Cluster III at 0.84. The highest cluster mean for pulp weight was found in Cluster IV (2.10), with Cluster II following at 1.29, and the lowest mean observed in Cluster III (0.63). Regarding pulp percentage, Cluster IV had the highest mean value at 84.69%, with Cluster V close behind at 83.12%, while Cluster III had the lowest mean value at 74.46%. Shell thickness had the highest cluster mean values in Cluster IV (3.37) and Cluster II (3.03), with the lowest mean recorded in Cluster I (2.29). The highest cluster mean for fruit circumference was found in Cluster IV at 54.18, followed by Cluster I at 43.36, while Cluster III had the lowest mean at 30.49. For fruit width, the maximum cluster mean of 17.24 was observed in Cluster IV, with Cluster I at 13.80 and the minimum mean in Cluster III at 9.70. Fruit length had the highest cluster mean in Cluster IV at 19.51, followed by Cluster VI at 15.84, with the lowest mean observed in Cluster III at 11.72. The number of seeds per fruit showed the highest cluster mean in Cluster IV at 211.74, followed by Cluster VI at 165.98, while Cluster III recorded the lowest mean at 96.65 [3,52].

The total seed weight was highest in Cluster IV (34.14) and Cluster I (29.44), with the lowest mean observed in Cluster III (22.18). The highest cluster means for fiber weight was noted in Cluster I (46.82) and Cluster II (44.94), whereas Cluster VI recorded the lowest value at 38.34. Cluster III had the highest mean for seed percentage at 2.72, followed by Cluster I at 2.31, while Cluster IV had the lowest mean at 1.38. The shell percentage was highest in Cluster III (18.06) and Cluster I (17.26), with the lowest mean observed in Cluster V (12.18). The highest cluster means for TSS pulp

**Table 6. Cluster mean analysis for 20 traits in beal genotypes.**

| S.N | Character | Clusters | | | | | |
| --- | --- | --- | --- | --- | --- | --- | --- |
| | | I | II | III | IV | V | VI |
| 1 | Shell weight | 222.51 | 256.37 | 149.42 | 304.51 | 179.68 | 198.13 |
| 2 | Fruit weight | 1.30 | 1.62 | 0.84 | 2.48 | 1.49 | 1.50 |
| 3 | Pulp weight | 1.00 | 1.29 | 0.63 | 2.10 | 1.24 | 1.23 |
| 4 | Pulp | 76.74 | 79.57 | 74.46 | 84.69 | 83.12 | 82.29 |
| 5 | Shell thickness | 2.29 | 3.03 | 2.43 | 3.37 | 2.34 | 2.42 |
| 6 | Fruit circumference | 43.36 | 42.09 | 30.49 | 54.18 | 41.17 | 42.00 |
| 7 | Fruit width | 13.80 | 13.39 | 9.70 | 17.24 | 13.10 | 13.36 |
| 8 | Fruit length | 15.53 | 15.58 | 11.72 | 19.51 | 15.42 | 15.84 |
| 9 | No. of seeds/fruit | 148.20 | 148.64 | 96.65 | 211.74 | 138.69 | 165.98 |
| 10 | Total seed weight | 29.44 | 27.02 | 22.18 | 34.14 | 26.62 | 29.15 |
| 11 | Fiber weight | 46.82 | 44.94 | 39.80 | 40.30 | 42.33 | 38.34 |
| 12 | Seed | 2.31 | 1.68 | 2.72 | 1.38 | 1.81 | 1.94 |
| 13 | Shell | 17.26 | 15.96 | 18.06 | 12.30 | 12.18 | 13.21 |
| 14 | TSS Pulp | 40.20 | 38.03 | 37.65 | 38.79 | 37.90 | 40.55 |
| 15 | Vitamin C | 17.99 | 17.46 | 18.63 | 18.19 | 19.75 | 23.35 |
| 16 | Acidity | 0.55 | 0.58 | 0.57 | 0.54 | 0.54 | 0.56 |
| 17 | Total sugar | 20.30 | 20.93 | 20.67 | 20.13 | 20.08 | 20.65 |
| 18 | Phenolic content | 20.25 | 28.33 | 29.95 | 26.19 | 29.07 | 30.62 |
| 19 | Non Reducing sugar | 17.04 | 17.77 | 17.53 | 17.04 | 16.68 | 17.16 |
| 20 | Fruit yield/plant | 93.34 | 103.96 | 75.45 | 108.99 | 99.73 | 104.56 |

were found in Cluster VI (40.55) and Cluster I (40.20), while Cluster III had the lowest mean at 37.65. Vitamin C levels were highest in Cluster VI (23.35), followed by Cluster V (19.75), with the lowest value in Cluster II (17.46). The highest cluster mean for acidity was recorded in Cluster II (0.58) and Cluster III (0.57), with the lowest mean in Cluster IV and Cluster V (0.54). For total sugar, the maximum cluster mean was observed in Cluster II (20.93), followed by Cluster III (20.67), while the minimum was recorded in Cluster V (20.08). Phenolic content was highest in Cluster VI (30.62) and Cluster III (29.95), with the lowest mean in Cluster I (20.25). Non-reducing sugar was highest in Cluster II (17.77), followed by Cluster III (17.53), and the lowest in Cluster V (16.68). Finally, the highest cluster mean for fruit yield per plant was observed in Cluster IV (108.99) and Cluster VI (104.56), with the lowest mean in Cluster III (75.45) [3,52]. Genotypes with the highest average values demonstrate significant potential for selection. Consequently, it is essential to prioritize Cluster IV for selecting inbred varieties and developing new cultivars in future breeding programs.

### Regression analysis

The OLS regression results indicate a model that explains a significant proportion of the variance in fruit yield per plant (Table 7). However, many individual predictors do not show statistically significant effects on the yield. Potential issues with multicollinearity might be present, affecting the coefficient estimates' reliability. The model seems overall robust but

**Table 7. Values of partial regression coefficient for growth and quality traits in beal genotypes.**

| S.N | Character | Coef | Std error | t | P > t | [0.025 | 0.975] |
|---|---|---|---|---|---|---|---|
| 1 | Shell weight | −248.213 | 236.948 | −1.048 | 0.298 | −719.578 | 223.152 |
| 2 | Fruit weight | 0.012 | 0.094 | 0.129 | 0.898 | −0.175 | 0.199 |
| 3 | Pulp weight | −0.483 | 5.066 | −0.095 | 0.924 | −10.561 | 9.595 |
| 4 | Pulp | −0.484 | 5.066 | −0.096 | 0.924 | −10.562 | 9.594 |
| 5 | Shell thickness | 3.474 | 2.346 | 1.480 | 0.143 | −1.194 | 8.141 |
| 6 | Fruit circumference | −1.281 | 1.931 | −0.663 | 0.509 | −5.122 | 2.561 |
| 7 | Fruit width | −158.811 | 204.163 | −0.778 | 0.439 | −564.956 | 247.333 |
| 8 | Fruit length | 499.991 | 641.462 | −0.779 | 0.438 | −776.080 | 1776.062 |
| 9 | No. of seeds/fruit | −1.043 | 0.828 | −1.260 | 0.211 | −2.690 | 0.604 |
| 10 | Total seed weight | 0.031 | 0.043 | 0.717 | 0.476 | −0.055 | 0.118 |
| 11 | Fiber weight | 0.579 | 0.602 | 0.963 | 0.338 | −0.617 | 1.776 |
| 12 | Seed | 0.306 | 0.179 | 1.716 | 0.090 | −0.049 | 0.662 |
| 13 | Shell | −7.538 | 5.511 | −1.368 | 0.175 | −18.502 | 3.425 |
| 14 | TSS pulp | 3.387 | 3.437 | 0.985 | 0.327 | −3.450 | 10.224 |
| 15 | Vitamin C | −0.169 | 0.273 | −0.618 | 0.538 | −0.712 | 0.374 |
| 16 | Acidity | 0.215 | 0.246 | 0.872 | 0.386 | −0.275 | 0.704 |
| 17 | Total sugar | 0.075 | 0.130 | 0.582 | 0.562 | −0.182 | 0.333 |
| 18 | Phenolic content | 1.866 | 1.209 | 1.543 | 0.127 | −0.540 | 4.271 |
| 19 | Non reducing sugar | −0.069 | 0.137 | −0.505 | 0.615 | −0.314 | 0.203 |
| 20 | Fruit yield/plant | −1.690 | 1.161 | −1.456 | 0.149 | −3.999 | 0.619 |

| | |
|---|---|
| Dep. variable: fruit yield/plant (kg) | Log-Likelihood:-329.31 |
| R-squared: 0.749 | Adj. R-squared: 0.694 |
| F-statistic: 13.58 | Prob (F-statistic): 1.20e-17 |
| AIC: 696.6 | BIC: 746.3 |
| Omnibus: 1.715 | Prob (Omnibus): 0.424 |
| Durbin-Watson: 1.928 | Cond. No: 371e+18 |
| Jarque-Bera(JB): 1.229 | Prob(JB): 0.541 |
| Skew: −0.051 | Kurtosis: 3.531 |

may benefit from further investigation into multicollinearity and possibly refining the predictor set. This table summarizes the relationship between several predictor variables and the dependent variable, Fruit yield/plant (kg). The model explains approximately 74.9% of the variability in fruit yield per plant. This relatively high value suggests that the model explains a significant portion of the variance. The F-statistic tests present the overall significance of the model, where a high value indicates that the model significantly improves prediction compared with a model with no predictors.

Similarly, a very low p-value of the Prob F-statistic test indicates that the overall model is statistically significant. A positive coefficient (pulp%, fruit width, number of seeds/fruits, total seed weight, fiber weight, shell%, vitamin C, acidity%, and total sugars) indicates a positive relationship with the dependent variable, while a negative coefficient indicates a negative relationship. The total sugar content has a positive association with fruit yield. The lack of statistical significance across most traits suggests that further research is needed to identify the key factors influencing yield in bael genotypes. The overall model fit is moderate, with an R-squared value of 0.749, indicating that while the model can explain some variance in yield, many traits may not significantly impact fruit yield. Fig 19 represents the association between fiber weight (g) and "Fruit yield/plant (kg) based on the OLS regression model. The red line shows the regression line, whilst the blue crosses illustrate the simulated data points. This image shows the correlation between fiber weight (g) and Fruit yield/plant (kg) using the regression model coefficients.

## Conclusions

The present study highlights the significance of the regional and national bael gene pool in semi-arid conditions by examining variability in morphometric, yield, and quality traits, with a particular emphasis on yield and qualitative attributes. Notably, high phenotypic coefficient of variation (PCV) and genotypic coefficients of variation (GCV) were observed for shell weight, fruit weight, and pulp weight, demonstrating substantial genetic diversity within the germplasm and a strong potential for selective breeding [3,16,17,19,21,23,25,37,48,58]. The heritability of traits varied significantly, from a low of 0.07% for fruit weight to a high of 92.23% for shell weight, indicating varying influences of environmental factors. Principal Component Analysis (PCA) revealed that the first principal component explained about 40.19% of the total variation, with a high eigenvalue of 8.12. The first six principal components collectively explained about 80.77% of the total variability, highlighting their significance in capturing the traits influencing bael phenotypes.

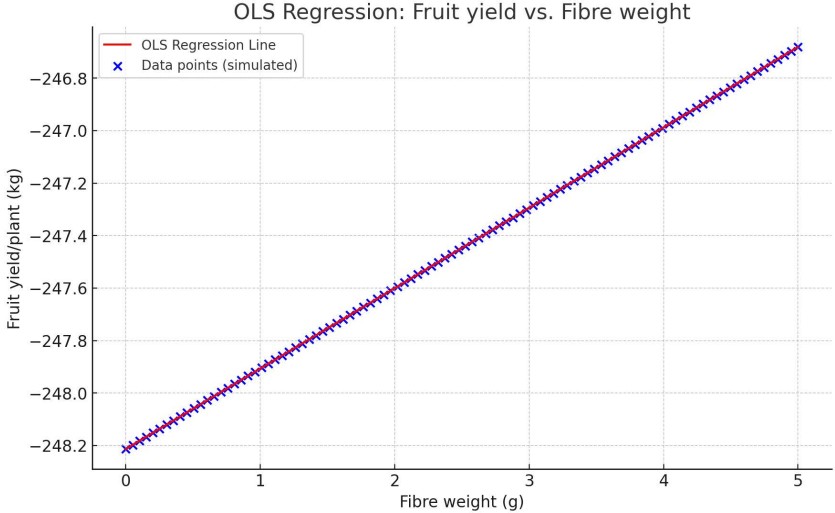

**Fig 19. The association between fiber weight (g) and fruit yield/plant (kg) based on the OLS regression model in bael genotypes.**

The highest positive PC scores for the first and second principal components were observed in genotypes CHESB-25 and CHESB-29, indicating their superior performance in these components. Additionally, based on their performance, CHESB-1, CHESB-8, CHESB-19, CHESB-27, CHESB-34, CHESB-39, CHESB-41, CHESB-45, CHESB-50, CHESB-52, CHESB-53, CHESB-56, CHESB-64, CHESB-71, CHESB-73, CHESB-81, CHESB-83, CHESB-85, CHESB-93, CHESB-95, and CHESB-98 also recorded good scores based on their performance, which could be useful for planning and carrying out breeding programs. Cluster analysis identified six distinct clusters, with Cluster V being the largest and Cluster VI the smallest. This clustering underscores the genetic diversity among bael genotypes, which can inform breeding and selection strategies. The positive correlations of fruit yield per plant with various traits such as shell weight, fruit weight, and pulp weight emphasize the importance of focusing on these traits to enhance yield. Cluster analysis reveals substantial variation in growth and yield traits across bael genotypes, with Cluster IV consistently showing the highest values for key attributes like shell weight, fruit weight, and yield per plant. Prioritizing Cluster IV is recommended for selecting superior inbred varieties and developing new cultivars in breeding programs. This research provides valuable insights into the extensive natural variation among 101 bael genotypes, emphasizing the significance of identifying elite genotypes within a large gene pool. Such insights are crucial for advancing genetic improvement efforts on both national and global scales.

## Supporting information

**S1 File. Raw data of the studied germplasm of bael.**
(XLSX)

## Author contributions

**Conceptualization:** A. K. Singh, Vikas Yadav, Lalu Prasad Yadav.

**Data curation:** Vikas Yadav, Lalu Prasad Yadav, K. Gangadhara, Anil Pawar.

**Formal analysis:** K. Gangadhara, Anil Pawar.

**Methodology:** A. K. Singh, Jagadish Rane, P. Ravat, A. Sahil, Prashant Kaushik, Ali Khadivi, Yazgan Tunç.

**Visualization:** A. K. Singh, Vikas Yadav, Lalu Prasad Yadav, Jagadish Rane, P. Ravat, A. Sahil, Prashant Kaushik, Ali Khadivi, Yazgan Tunç.

**Writing – original draft:** Vikas Yadav, Lalu Prasad Yadav.

**Writing – review & editing:** A. K. Singh, K. Gangadhara, Anil Pawar, Jagadish Rane, P. Ravat, A. Sahil, Prashant Kaushik, Ali Khadivi, Yazgan Tunç.

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
