## [Decision Letter · Decision Letter 0]

22 Feb 2026

Dear Dr. Khadivi,

Thank you for submitting your manuscript to PLOS ONE. After careful consideration, we feel that it has merit but does not fully meet PLOS ONE’s publication criteria as it currently stands. Therefore, we invite you to submit a revised version of the manuscript that addresses the points raised during the review process.

**ACADEMIC EDITOR:** The manuscript must be subjected to considerable rearrangements and thorough revising. Check the manuscript properly and remove the grammatical mistakes. Also check all references. Some references did not match the journal reference style.The manuscript must be subjected to considerable rearrangements and thorough revising. Check the manuscript properly and remove the grammatical mistakes. Also check all references. Some references did not match the journal reference style.

We look forward to receiving your revised manuscript.

Kind regards,

Alia Ahmed

Academic Editor

PLOS One

Journal Requirements:

3. We note that your Data Availability Statement is currently as follows: “All relevant data are within the manuscript and its Supporting Information files.”

4. We note that Figure 1 in your submission contain map images which may be copyrighted. All PLOS content is published under the Creative Commons Attribution License (CC BY 4.0), which means that the manuscript, images, and Supporting Information files will be freely available online, and any third party is permitted to access, download, copy, distribute, and use these materials in any way, even commercially, with proper attribution. For these reasons, we cannot publish previously copyrighted maps or satellite images created using proprietary data, such as Google software (Google Maps, Street View, and Earth). For more information, see our copyright guidelines: http://journals.plos.org/plosone/s/licenses-and-copyright.

1. You may seek permission from the original copyright holder of Figure(s) [#] to publish the content specifically under the CC BY 4.0 license.

5. We note that Figures 2 - 13 in your submission contain copyrighted images. All PLOS content is published under the Creative Commons Attribution License (CC BY 4.0), which means that the manuscript, images, and Supporting Information files will be freely available online, and any third party is permitted to access, download, copy, distribute, and use these materials in any way, even commercially, with proper attribution. For more information, see our copyright guidelines: http://journals.plos.org/plosone/s/licenses-and-copyright.

1. You may seek permission from the original copyright holder of Figure(s) [#] to publish the content specifically under the CC BY 4.0 license.

Reviewers' comments:

Reviewer's Responses to Questions

**Comments to the Author**

1. Is the manuscript technically sound, and do the data support the conclusions?

Reviewer #1: Yes

Reviewer #2: Yes

2. Has the statistical analysis been performed appropriately and rigorously?

Reviewer #1: Yes

Reviewer #2: Yes

3. Have the authors made all data underlying the findings in their manuscript fully available?

Reviewer #1: Yes

Reviewer #2: Yes

4. Is the manuscript presented in an intelligible fashion and written in standard English?

Reviewer #1: Yes

Reviewer #2: Yes

Reviewer #1: The manuscript presents a valuable phenotypic and multivariate assessment of a large Aegle marmelos germplasm collection. However, minor revision is needed to address language quality, redundancy, and over-interpretation of results as suggested in reviewed MS. With careful grammatical editing and incorporation of suggestions, the MS may be accepted for publication.

Reviewer #2: The research article is fit for publication in its present form. The findings have been explained using standard statistical tools. Sufficient number of accessions have been included for drawing conclusion.

.

Reviewer #1: No

Reviewer #2: **Yes:** Dr. Amit Jasrotia, Professor (Fruit Science), Sher-e-Kashmir University of Agricultural Sciences & Technology-Jammu, Jammu & Kashmir, INDIA-180009.Dr. Amit Jasrotia, Professor (Fruit Science), Sher-e-Kashmir University of Agricultural Sciences & Technology-Jammu, Jammu & Kashmir, INDIA-180009.Dr. Amit Jasrotia, Professor (Fruit Science), Sher-e-Kashmir University of Agricultural Sciences & Technology-Jammu, Jammu & Kashmir, INDIA-180009.Dr. Amit Jasrotia, Professor (Fruit Science), Sher-e-Kashmir University of Agricultural Sciences & Technology-Jammu, Jammu & Kashmir, INDIA-180009.

---

## [Author Response · Author response to Decision Letter 1]

26 Mar 2026

Dear Editor,

Thank you for your comments.

All requested corrections have been carefully addressed. In particular, all figures have been uploaded as separate high-resolution files in accordance with the PLOS ONE guidelines.

Kind regards,

Prof. Dr. Ali Khadivi

(On behalf of all authors)

---

## [Editor Report · Decision Letter 1]

8 Apr 2026

Phenotypic variability and trait-specific selection in Aegle marmelos Correa genotypes based on morphological and quality traits

PONE-D-26-04072R1

Dear Dr. Khadiviv,

We’re pleased to inform you that your manuscript has been judged scientifically suitable for publication and will be formally accepted for publication once it meets all outstanding technical requirements.

Kind regards,

Alia Ahmed

Academic Editor

PLOS One
---

## [Editor Report · Acceptance letter]

PONE-D-26-04072R1

PLOS One

Dear Dr. Khadivi,

I'm pleased to inform you that your manuscript has been deemed suitable for publication in PLOS One. Congratulations! Your manuscript is now being handed over to our production team.

Kind regards,

on behalf of

Dr. Alia Ahmed

Academic Editor

PLOS One